# e3: Learning to Explore Enables Extrapolation of Test-Time Compute for LLMs

**Amrith Setlur**[1,*], **Matthew Y. R. Yang**[1,*], **Charlie Snell**[2], **Jeremy Greer**[3], **Ian Wu**[1],
**Virginia Smith**[1], **Max Simchowitz**[1], **Aviral Kumar**[1]
[1]Carnegie Mellon University, [2]UC Berkeley, [3]Oumi

## Abstract

Test-time scaling offers a promising path to improve LLM reasoning by utilizing more inference compute; however, the true promise of this paradigm lies in *extrapolation* (*i.e.*, improvement in performance as LLMs keep "thinking" for longer, far beyond the maximum token budget they were trained on). Surprisingly, we find that most existing, open-source reasoning models do not extrapolate well. We show that one way to enable extrapolation is by training the LLM to perform *in-context exploration*: spending token budget on chaining operations (generation, verification, summarization, *etc.*) or testing multiple hypotheses before committing to an answer. To enable in-context exploration, we identify three key ingredients as part of our recipe e3: **(1)** chaining skills that the base LLM has asymmetric competence in, *e.g.*, chaining verification (easy) with generation (hard), as a way to implement in-context search; **(2)** leveraging "negative" gradients from incorrect traces to amplify in-context exploration during RL, resulting in longer traces chaining additional asymmetries; and **(3)** coupling task difficulty with training token budget via a specifically-designed RL curriculum to structure this in-context exploration. Our recipe e3 produces one of the best known 1.7B models on AIME/HMMT'25, and extrapolates to 2.5× the training token budget. Our e3-1.7B not only performs well, but it can go beyond the problem-solving capabilities of the base model itself.

## 1 Introduction

Test-time scaling boosts large language model (LLM) performance by spending more compute on "thinking" before producing an answer. Its ultimate promise lies in enabling models to continue improving performance by scaling test compute at deployment. *E.g.*, if the model can learn to implement "algorithmic procedures" like planning, self-reflection, or backtracking, it can discover more accurate responses as more test compute is used. With this motivation, current recipes post-train LLMs via reinforcement learning (RL) (DeepSeek-AI et al., 2025; Yu et al., 2025) and supervised fine-tuning (SFT) (Team, 2025; Muennighoff et al., 2025) at long context windows. However, it is unclear whether these new models can truly realize the promise of *extrapolation*: if we scale test compute beyond the maximum *training budget*, would the LLM be able to solve more problems?

Although performance at very long response lengths may be restricted by other factors like model architecture or context lengths (Li et al., 2024), one can at least expect that an LLM should benefit from test-time scaling within the pre-training and fine-tuning training budget. Mechanistically, this can be realized by implementing algorithmic procedures (*e.g.,* generate-verify-revise, best-of-$N$) within the model's chain of thought, in-context (Kumar et al., 2024; Setlur et al., 2025a; Gandhi et al., 2024). However, similar to other empirical studies of reasoning models, we note that many open models perform poorly when extrapolating to 2-3× the training budget (Qu et al., 2025b; Hochlehnert et al., 2025). Thus, relying on current RL/SFT recipes to yield effective extrapolation is mostly futile.

In this paper, we show that the key to enabling extrapolation is *learning to explore in-context*: if a model learns to use compute by searching through multiple reasoning paths or implementing algorithmic procedures, it can "guide" the search towards the correct answer, and improve its performance as more test compute becomes available. Even under the original training compute budget, we expect learning to explore in-context to improve performance on unseen, out-of-distribution problems (Ghosh et al., 2021; Duan et al., 2016). To demonstrate this, we build a recipe e3, which

---

*Equal contribution. Correspondence to asetlur@andrew.cmu.edu. This work was done at CMU. Project website: https://matthewyryang.github.io/e3/.

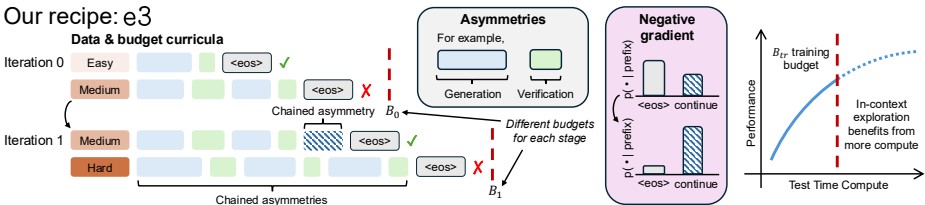

Figure 1: *In-context exploration enables extrapolation* (`e3`): (i) chaining asymmetric capabilities in the base model, *e.g.,* reliably self-verifying responses after generating them; (ii) using negative gradients in RL training to penalize incorrectly terminated model responses, lengthening them further with more chained asymmetries, until the correct answer is discovered; and (iii) data & budget curricula for RL training that carefully balances explore-exploit tradeoff by sequentially training models on different datasets and training compute budgets. Qwen3-1.7B fine-tuned with `e3` extrapolates test-time compute outperforming all ≤2B models on AIME'25.

trains models to explore in-context so that they can perform well at both the training and extrapolation budgets. At its core, `e3` is based on the following three ingredients (see Fig. 1):

**1) Asymmetries critical for learning for explore.** LLMs can learn to explore only when each segment in the output trace is useful in "guiding" subsequent ones, *e.g.*, if verifying initial segments can lead to more refined answers later on. In the absence of external tools, we show that feedback can emerge from *asymmetries*, which are differences in the model's competence at different procedures that constitute an output trace. In the context of self-verification, this corresponds to a verification-generation (VG) gap, where models are more capable of verifying their answers than generating correct ones. While prior work (Setlur et al., 2025b; Swamy et al., 2025; Song et al., 2024; Kim et al., 2025; Gandhi et al., 2025) has noticed such asymmetries, we show that these are critical for extrapolation, meaning that in their absence, scaling is strikingly hard. We conceptualize this by building a didactic model of in-context exploration under the generation-verification asymmetry, termed $p^k$ and prove theoretically that chaining asymmetries can drive exploration.

**2) Negative gradient in RL amplifies in-context exploration.** If asymmetries are a prerequisite for learning to explore, what mechanism gets the LLM to chain them more and facilitate useful exploration strategies during RL? We show that *negative gradients* (*i.e.*, gradients on incorrect traces) in RL is a key enabler of in-context exploration, when the base model presents asymmetries. Negative gradients drive exploration (Tajwar et al., ICML 2024; Ren & Sutherland, 2024) by moving the probability mass from shorter failed traces onto longer traces that *"chain"* new asymmetries (*e.g.*, LLM verifying a calculation one more time). In contrast, SFT only maximizes likelihood on correct traces in the training data and reinforces the model to end the solution within the length of these traces. Conceptually, SFT only aims to reduce the failure probability $p$ at a fixed $k$ in our $p^k$ model, whereas negative gradients also amplify $k$, which is the number of attempts made in-context.

**3) Structured exploration with coupled curriculum.** Finally, while negative gradients amplify asymmetries and produce longer responses, running RL training at longer budgets suffers from poor training convergence, typically seen in long-horizon RL (Agarwal et al., 2021). While one could train with a smaller budget, we show that training on hard problems at short context lengths often disincentivizes exploration altogether since the model is forced to commit to an answer prematurely. To resolve this, we design a *coupled curriculum* over pairs of (data mixture, training budget) that effectively structures the exploration driven by the negative gradient. Our key insight is that at any stage of the curriculum, we choose the smallest "RL optimization friendly" budget such that the model initialized for RL training can: (i) complete most of its responses within the budget; and (ii) can continue to improve performance as it chains more asymmetries beyond the chosen budget.

The above insights constitute our recipe `e3`, that we use to post-train the Qwen3-1.7B model with a training budget of up to 16k output tokens. We build the ***one of the best performing reasoning models at <2B scale on AIME'25 and HMMT'25*** (to our knowledge), and our model consistently improves as we extrapolate compute to 32k (2× training budget). `e3`-1.7B also attains consistent improvements on the pass@32 metric, showing that `e3` does more than simply sharpening the base model. `e3`-1.7B also improves on non-math reasoning benchmarks (see App. K), not trained upon.

## 2 RELATED WORK

**Scaling test-time compute via long CoT.** Recent work scaling test-time compute by training models to generate long chains of thought (CoT) that combine verification (Zhao et al., 2025a), search (Lu

et al., 2025), and self-correction (Kumar et al., 2024) has achieved SOTA performance on various reasoning benchmarks (DeepSeek-AI et al., 2025; Team et al., 2025; OpenAI et al., 2024), resulting in widespread open-source efforts (Face, 2025; Yeo et al., 2025; Zeng et al., 2025b; Luo et al., 2025b). The true benefit of test-time scaling is performance gains under extrapolation. Prior work prompts models to generate extra tokens when a response ends (Muennighoff et al., 2025; Aggarwal & Welleck, 2025), but we show models that can explore in-context well, extrapolate more effectively.

**Exploration in test-time scaling.** Long CoTs allow models to explore various approaches before committing to a final answer. While prior works show the role of exploration in reasoning (Gandhi et al., 2025; Liu et al., 2025b; Krishnamurthy et al., 2024; Nie et al., 2024), we discover the crucial enabling factor is the presence of asymmetries, like verification-generation gap (Setlur et al., 2025a; Song et al., 2024; Swamy et al., 2025), in the model. Moreover, while concurrent work builds techniques to boost exploration during RL via advantage normalization (Li et al., 2022; Yu et al., 2025; Zeng et al., 2025a) or PPO clipping (Yu et al., 2025), they do not highlight the role of negative gradients which as we show incentivizes the chaining of asymmetries, and scales up in-context exploration. Concurrent works (Wang et al., 2025b; Zhu et al., 2025) remark about the role of negative gradient and discuss entropy trends when running RL. Our study formally investigates the underlying mechanism of negative gradients increasing length and entropy, with theoretical results.

**RL training with data and length curricula.** Recent works investigate curriculum training over problem difficulty (Team et al., 2025; Xie et al., 2025; Shi et al., 2025), going from easy to hard, and over training budget (Luo et al., 2025b; Liu et al., 2025a), from short to long, during RL. Their primary motivation is efficiency: avoiding zero-advantage updates (Shi et al., 2025; Yu et al., 2025), efficient optimization (Luo et al., 2025b), or efficiency of using test-time compute (Qu et al., 2025b).While we observe similar trends for each curriculum, our key finding is that coupling data and budget curricula enables in-context exploration and improves extrapolation performance beyond mere compute efficiency. We show that training on hard problems with short budgets often yields terse solutions that fail to generalize, while easy problems with long budgets can produce overly verbose outputs (Sec. 6). See App. C for detailed discussion of related works.

# 3 PROBLEM STATEMENT: OPTIMIZING & EXTRAPOLATING TEST COMPUTE

**Problem statement.** RL and SFT are categories of post-training algorithms that refine a pre-trained base LLM $\pi_b$ into a reasoning model, in particular one that produces long chains-of-thought to succeed. Typical outcome-reward RL trains LLM $\pi$ (initialized with $\pi_b$) to maximize performance on outcome 0/1 reward $r^\star(\mathbf{x}, \mathbf{y})$, for inputs $\mathbf{x} \sim \rho$ and response $\mathbf{y} \sim \pi(\mathbf{y}|\mathbf{x})$ restricted to an apriori fixed maximum token length or *training budget* $B_{\mathrm{tr}}$ (Yu et al., 2025; Luo et al., 2025b). On the other hand, SFT fine-tunes $\pi_b$ on long thinking traces from more capable models or humans to distill their reasoning capabilities (Team, 2025; Muennighoff et al., 2025), where the maximum length of the expert traces also implicitly induces a training budget $B_{\mathrm{tr}}$, similar to RL. **Our goal**, is to train models that can improve performance when we extrapolate test-compute beyond $B_{\mathrm{tr}}$. Even though the true promise of test-time compute is extrapolation performance, we find that *current thinking models fall short on extrapolation.* We evaluate

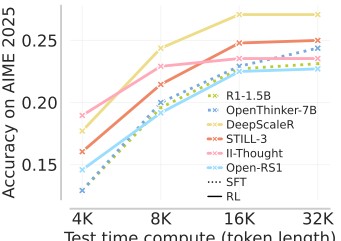

Figure 2: Accuracy on AIME 2025 of various open-source models at different test time compute budgets. Performance gains diminish as the test-time budget increases, with virtually no gains from 16k to 32k.

multiple models on a test budget of 32K, $\approx 1.5\text{-}2\times B_{\mathrm{tr}}$ across all models, on AIME25 in Fig. 2 (see App. B for a detailed comparison) and note that most of the performance gains lie in the training budget, and the gains are minuscule as we test beyond that.

**Negative gradient in RL.** A key distinction between SFT and RL is the *negative gradient*, which corresponds to the part of the policy gradient coming from traces that fail. In Eq. 1 we present a generalized version of the policy gradient adopted by most RL post-training methods. From this we note that on a prompt $\mathbf{x}$, RL training observes two types of gradients: (i) the positive gradient which maximizes the likelihood of a correct responses $\mathbf{y}$ with a positive advantage $A(\mathbf{x}, \mathbf{y})$, and (ii) the negative gradient which *pushes down* the likelihood of an incorrect response with a negative advantage $A(\mathbf{x}, \mathbf{y})$. Here, $\mathbf{y}$ can be sampled *on-policy* $\pi = \tilde{\pi}$ or *off-policy* $\pi \neq \tilde{\pi}$. Thus, we can view SFT as a purely positive gradient method that only maximizes likelihood on correct reasoning traces. In Sec. 5, we show why the negative gradient is largely responsible for driving up response lengths

and in-context exploration during RL, resulting in better extrapolation than SFT.

$$\mathbb{E}_{\mathbf{y} \sim \tilde{\pi}(\cdot | \mathbf{x})} \left[ A_i(\mathbf{x}, \mathbf{y}) \cdot \nabla_\pi \log \pi(\mathbf{y} | \mathbf{x}) \right] \quad \text{(general form of policy gradient in RL)} \quad (1)$$

## 4 CHAINING ASYMMETRIES: PRE-REQ. FOR IN-CONTEXT EXPLORATION

*How can extrapolating beyond the training budget improve performance?* We begin by revisiting why longer traces perform better in general. The conventional wisdom is that longer traces can represent solutions that make multiple attempts, interleaving verification and generation (Setlur et al., 2025b; Nie et al., 2024; Krishnamurthy et al., 2024), to arrive at the final answer. We can think of this as the LLM learning to interleave basic "skills", *e.g.*, verification, summarization, or general instruction following, to perform in-context exploration. But why, or when, should post-training favor such traces over other shorter ones that arrive at the answer directly? We demonstrate that when the base model exhibits *asymmetric* incompetence at different skills, RL post-training prefers to learn solutions that *chains asymmetric skills* in ways that improve final performance. A formal description is:

> **Definition 4.1** (Chaining asymmetric capabilities $p, q$ in model $\pi$.). Let $p, q : \mathcal{S} \mapsto \mathcal{S}$ be functions over token sequences $\mathcal{S}$ (*e.g.*, $p$ can be generation, $q$ can be verification), and $\texttt{detect}(f, \tau)$ detects number of calls to function $f$ in a token trace $\tau$. For a reward $r$, we say that policy $\pi$ chains asymmetries $p, q$ if it benefits from calls to the composition $q(p(\cdot))$, compared to only $p(\cdot)$:
> $$\mathbb{E}_{\tau \sim \pi} \left[ r(\tau) \mid \texttt{detect}(q(p(\cdot)), \tau) > 0 \right] > \mathbb{E}_{\tau \sim \pi} \left[ r(\tau) \mid \texttt{detect}(p, \tau) > 0 \right],$$
> even though there is an optimal policy $\pi_r^\star$ that never calls $q$, i.e., $\mathbb{E}_{\tau \sim \pi_r^\star} \left[ \texttt{detect}(q, \tau) \right] = 0$.

We focus on a key special case when the model is more accurate at verifying its own answers than it is at generating correct ones; that is, when the model exhibits a *verification-generation gap* (**VG Gap**), on a particular problem domain (Song et al., 2024; Setlur et al., 2025b; Swamy et al., 2025). In this section, we show that RL training on problem domains with VG gap (i) encourages chaining asymmetries, (ii) enables in-context exploration that (iii) discovers new solutions, often extrapolating to larger budgets and more difficult problem domains. In App. F, we also discuss another chained asymmetry: (1) generating abstract plans or hints, then (2) solving from a chosen hint, rather than jumping straight to a reasoning attempt, analyzing results from a concurrent work (Qu et al., 2025a).

**Analysis setup.** We validate the role of asymmetries in learning to explore by investigating two didactic tasks, on which Llama3.2-3B admits different VG gaps. First, the **Countdown game** (Yao et al., 2023; Gandhi et al., 2024) (CDOWN) requires converting a set of numbers into an equation that evaluates to the desired target. The base LLM is more effective at verifying whether a proposed equation evaluates to the target than searching over all possible equations to solve the task, and traces with more chained asymmetries are more performant. We confirm this by measuring our Def. 4.1 in Fig. 3: performance on traces with more chains is higher, and both the number of chains and performance get amplified during RL. Second, we study **n-digit multiplication** (MULT) in natural language, where the base model exhibits limited verification (see App. E). Addition-

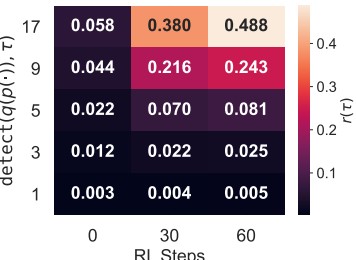

Figure 3: *Measuring asymmetry on* CDOWN. Rewards $r(\tau)$ improve as traces $\tau$ have more chains $q(p(\cdot))$ in them, and this is amplified during RL on Llama3.2-3B.

ally, we fine-tune Llama3.2-3B on correct n-digit multiplication traces from Qwen-32B-r1-distilled, which contains multiple intermediate steps verifying smaller digit multiplications (see App. E for an example). This fine-tuning is a direct way to encourage more verification attempts (MULT-V). Comparison of MULT vs. MULT-V enables direct evaluation of the benefits of asymmetries in base LLM, all else being held equal. In these results, we detect verification segments by looking for thought blocks that start with phrases like "Wait, ...", "Let me verify ...". (see App. N for examples).

*Finding 1: Verification-generation asymmetry in the base model improves the performance of RL trained solutions.* Fig. 4 (a,b) shows a stark difference in performance and length of output traces as the training budget $B_{\text{tr}}$ varies on CDOWN and MULT. On CDOWN, performance consistently increases as $B_{\text{tr}}$ increases from $512 \rightarrow 2048$, accompanied by a very clear increase in response length. On MULT, where the base model has limited propensity to verify, performance increases when $B_{\text{tr}}$ increases from 1024 to 2048, but it plateaus thereon. Unlike CDOWN, test-time length is far from saturating budget limits and also oscillates widely across RL training epochs. Contrast this with

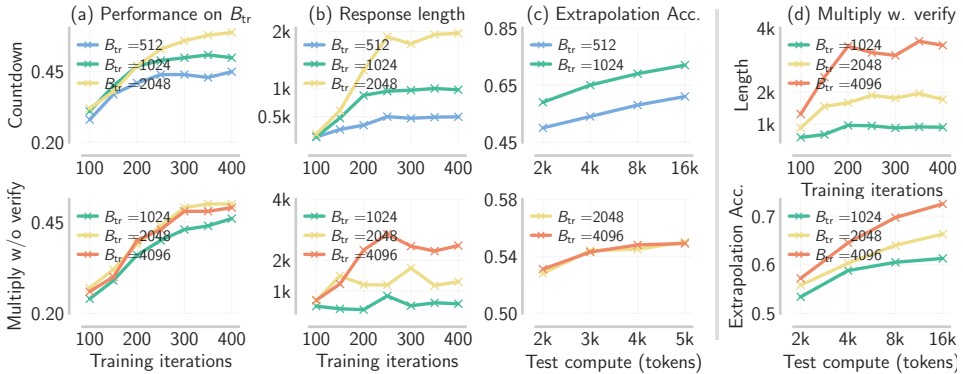

Figure 4: ***RL training with and without asymmetries in the base model.*** When asymmetries such as the VG gap are present (*e.g.,* in CDOWN), RL training amplifies response length by chaining more asymmetries to explore in-context, where the probability of success improves with higher length on both $B_{\text{tr}}$ and extrapolation regimes. On the other hand, when VG gap is absent in $\pi_b$ (*e.g.,* in MULT), increases in length and extrapolation performance are subdued. When we explicitly train on a base model fine-tuned to verify MULT (a setting we refer to as the MULT-V), we again observe upward length and extrapolation trends, consistent with CDOWN.

Fig. 4 (d), where RL training on MULT-V, which leverages verification, exhibits longer lengths and stronger extrapolation performance. Overall, this implies that **leveraging asymmetries improves performance and length-utilization during RL.** Curiously, we observe that models with greater VG gap exhibit less KL divergence from the base model, implying better generalization (see App. E).

*Finding 2: Chaining asymmetries enable extrapolation via in-context exploration.* Interleaving verification and generation steps chains together asymmetric skills of the base model; we refer to this special case of skill-chaining as *chaining asymmetries*. To measure the benefits of chained asymmetries on CDOWN, we plot the pass@k accuracy of the base LLM, shown in Figure 3, and observe that performance increases as more chained asymmetries arise. In fact, the best strategy is to not simply scale $k$, but rather to scale both $k$ and the number of chained asymmetries (details in App. E). In Fig. 4 (c), we plot the extrapolation performance of the models trained at two values of $B_{\text{tr}}$. On CDOWN the model trained with $B_{\text{tr}}$ 0.5-1k makes steady progress on problems in test budgets that are 8-16× $B_{\text{tr}}$ itself. On MULT, we find that $B_{\text{tr}}$ has absolutely no effect on extrapolation performance when the base LLM that does not have VG asymmetry, but it has a substantial effect when the asymmetry is present. More importantly, while the base model without VG asymmetry fails to extrapolate and solve unsolved problems, with its accuracy improving by merely $\leq 2\%$ despite 16× test-time compute scaling, the base model with VG asymmetry can still extrapolate well.

**Why do asymmetries enable in-context exploration?** We explain this using a didactic $p^k$-model (App. A), where we view the LLM as making guesses $a_1, \ldots, a_k$, each failing with probability $p$, under perfect verification. In this simplified case, failure probability $p^k$ decays exponentially, so performance improves by increasing both $k$ and $p$. But if verification is hard, more guesses give little benefit, and gains come only from lowering $p$ (better first guesses, as in MULT).

> **Takeaways: Asymmetries are a critical pre-requisite for learning to explore.**
>
> • Asymmetries like the VG gap enable the model to verify and refine answers in-context.
> • RL amplifies chaining of asymmetric skills in base model and produces solutions that learn to explore in-context, benefiting from more extrapolated compute at test-time.

## 5 NEGATIVE GRADIENTS DURING RL CHAIN MORE ASYMMETRIES

Given that asymmetries in the base model are a prerequisite for in-context exploration, we now ask: what enables models to exploit and chain these asymmetries during RL? We show that a crucial ingredient is the ***negative gradient***, the gradient term multiplied by a negative advantage in Eq. 1. Negative gradient drives in-context exploration via two mechanisms: (i) incentivizing sampling of unseen token sequences; (ii) chaining asymmetries like VG gap that rapidly drives up response length and in-context verification attempts. Note that while mechanism **(i)** corresponds to the *classical* notion of exploration, mechanism **(ii)** is special in that it corresponds to a form of "structured" exploration over strategies in the base model (corresponds to "meta exploration" (Liu et al., 2020) in RL terms). Below, we study these empirically, and in App. A we study it theoretically.

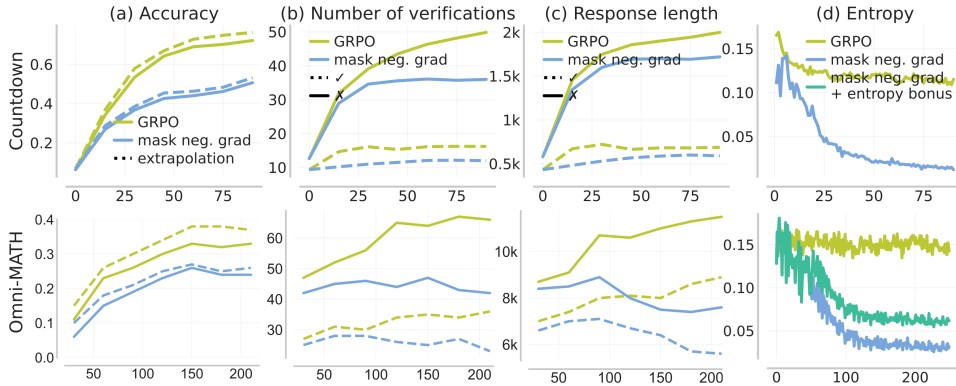

Figure 5: **_RL training with and without negative gradients:_** When the base model admits asymmetries like verification generation gap, negative gradients promote in-context exploration by: i) increasing length (c) and chaining asymmetries, which shows up as more verification attempts (b); and ii) increasing token entropy and thus response diversity (d). This leads to better performance on the training budget and upon extrapolation (a). In (b, c), ✓ denotes the statistic computed on correct responses and ✗ on incorrect responses.

**Analysis setup.** We analyze the evolution of response length, performance, and the number of chained asymmetries, comparing: (i) standard outcome-reward RL using GRPO (Liu et al., 2025b); (ii) GRPOMask, which zeros out (i.e. masks) the negative gradient and whilst retaining the *positive* gradient, thereby resembling an approach close to online STaR (Zelikman et al., 2022) or RFT (Yuan et al., 2023). We conduct experiments on CDOWN and DMATH reasoning (problems sourced from DeepScaleR (Luo et al., 2025b)) that exhibit the VG asymmetry and make the following observations:

*Finding 1: Negative gradients promote diverse responses during RL training*, encouraging exploration at two levels: (i) within a rollout; and (ii) across different rollouts. For (i), we observe that removing the negative gradient results in an entropy collapse over the next-token distribution (Fig. 5 (d)). This curtails diversity and leads to responses with a repeating stream of tokens when extrapolating token budget (Fig. 6). For (ii), we measure the cumulative unique attempts on the CDOWN as we train the model (Fig. 6). We separate each rollout into attempts using "\n" and parse the

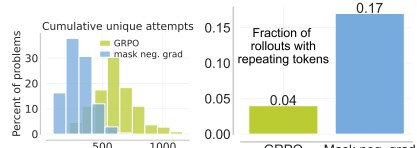

Figure 6: **_Negative gradient encourages distinct responses:_** increases the cumulative number of unique attempts on CDOWN (left) and reduces responses that end with a repeating stream of tokens on DMATH (right).

equations. An attempt is unique when its equations differ from other rollouts across training steps. We find more unique attempts when training with negative gradients, indicating better exploration. While it is not surprising that RL benefits from exploration (Hazan et al., 2019), distinctly from standard RL, this exploration can be particularly effective when extrapolating to test budgets (Fig. 5(a)).

*Finding 2: Negative gradient increases the number of chained asymmetries*, and thereby boosts structured exploration (and extrapolation). Concretely, when training on an incorrect response $\mathbf{y}$ with tokens $y_1, y_2, .., \text{EOS}$, the negative gradient reduces the conditional probability of each token $y_i$ conditioned on the prefix $y_{1:i-1}$ appearing in this response, i.e. $p(y_i|\mathbf{y}_{1:i-1})$. This process also reduces the probability of the EOS token: $p(\text{EOS}|\mathbf{y})$, for any incorrect response that ends within the response budget. Where does this probability mass go? Clearly since total probability is conserved, this probability mass must be repurposed to increase the likelihood of other tokens. Fig. 5(b) shows that the probability mass recovered from the negative gradient is repurposed to increase the probability of chaining new pairs of asymmetric skills to the current trace (*e.g.*, "Wait, ..." instead of terminating with EOS). This chaining results in a greater response length **(c)** and higher overall performance.

When negative gradients are masked (GRPOMask) in CDOWN or DMATH, verification attempts and length plateau or drop (Fig. 5(b, c)). The effect of masking is more pronounced for incorrect rollouts, but also holds similarly for correct rollouts, *i.e.*, the number of chained asymmetries and length plateaus or declines in the absence of negative gradients. We include more results in App. G (Fig. 16), where we also demonstrate that MULT (which does not exhibit asymmetries) benefits far less from negative gradients. This mechanism for boosting exploration by chaining new asymmetries is different from the typical notions of improving coverage or trying novel tokens discussed in Finding 1.

*Finding 3: LLMs trained with negative gradients extrapolate better.* Longer responses that chain asymmetries are more likely to yield correct answers and thus receive positive reward. Therefore,

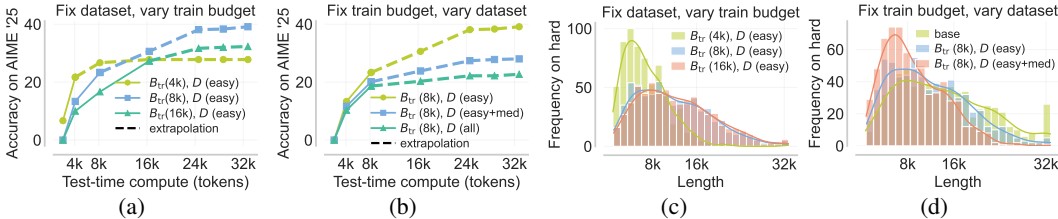

Figure 7: **RL training on different data/budgets. (a), (c):** Best results come from balancing optimization difficulty (better at shorter budgets) and in-context exploration (better at longer budgets). **(b), (d):** Training on hard problems at the 8k budget kills longer traces needed to discover solutions for hard problems

the policy gradient update reinforces chaining and improves in-context exploration, and this is reinforced: further training incentivizes more in-context exploration (since the gap between number of verifications with and without negative gradient increases as training progresses in Fig. 5(b)). As discussed in Sec. 1, models that learn to explore in-context benefit more from additional test-time compute. Fig. 5(a) confirms this: on hard DMATH problems (we classify a problem as hard if QwQ-32B attains pass@32 performance of *zero*), doubling the test-time budget amplifies the performance gap when negative gradients are used, compared to the masked variant.

> **Takeaways: Negative gradient in RL amplifies in-context exploration under large VG gaps**
>
> - Negative gradients in RL move the probability mass from a short-length incorrect trace onto a longer trace with more chained asymmetries that results in the correct answer.
> - Negative gradients boost response diversity and thus coverage over correct answers, as noted empirically on CDOWN, DMATH, and theoretically in our bi-gram model analysis (App. A).

## 6 COUPLED CURRICULUM STRUCTURES EXPLORATION IN LONG LENGTH RL

In the presence of asymmetries, training with negative gradients produces models that can extrapolate beyond their training budget. However, of course, training on just *any* arbitrarily chosen training token budget $B_{tr}$ is not enough: if $B_{tr}$ is too small, then we would not expect any form of in-context exploration to emerge. Perhaps unsurprisingly it turns out that a much larger $B_{tr}$ is also not sufficient. In Fig. 7(a), we show that different training budgets $B_{tr}$ lead to different levels of extrapolation performance. *So how should we set the budget $B_{tr}$ to attain strong extrapolation performance?* And, *what prompts should we be training on for a given budget?* We answer these questions below.

**Setup.** We evaluate extrapolation performance on DMATH and CDOWN after training on different budgets and prompt compositions. We split DMATH evenly across three levels of hardness as measured by the performance of Qwen-R1-Distilled-32B. For CDOWN, we judge problem difficulty based on the number of terms in the equation. We use the GRPO (Liu et al., 2025b) algorithm to train models on all compute budgets and datasets (see App. I for the hyperparameter configurations we use).

***Finding 1: Training solely at low or high $B_{tr}$ is not desirable.*** We train on the easy DMATH problems at different training budgets $B_{tr}$= 4k, 8k,16k (see Fig. 7(a)). While training at the short budget $B_{tr}$= 4k attains the best performance at the same test budget, it "kills" in-context exploration since traces with many chained asymmetries are typically longer than the training budget of 4k and only successes within 4k are rewarded. Overall, this hinders length increase and chaining of asymmetries driven by the negative gradient, leading to poor extrapolation (no gains from 8k to 32k). Fig. 7(c) shows that this biases the model to stop early and incorrectly. In contrast, training at $B_{tr}$= 16k introduces significant optimization challenges, stemming from policy gradient variance (Agarwal et al., 2021). This model performs worse on its own training budget of 16k compared to a model trained on $B_{tr}$= 8k and extrapolated to 16k. We find that $B_{tr}$= 8k attains the best extrapolation scaling.

***Finding 2: Training naïvely on a static data mixture is suboptimal.*** Having identified a reasonable training budget of 8k, we now turn to studying the effect of data compositions (prompt mixtures). To do so, we compare the naïve training data mixture with equal proportions of all difficulties (easy + medium + hard) against easy, easy + medium at $B_{tr}$= 8k. As expected, matching train and test composition is ideal for better *in-distribution* performance, *i.e.*, when evaluating models at a test budget of $B_{tr}$ (see App. I). However, perhaps surprisingly, the same is not true for performance on out-of-distribution (OOD) problems, especially when performance is computed at budgets $\gg B_{tr}$. As shown in Fig. 7(b), the model trained on *only easy* problems obtains the best performance on OOD AIME'25 when extrapolating compute to 32k. This is despite the fact that AIME'25 problems resemble hard ones and a few prior AIME problems are also present in the hard subset of DMATH.

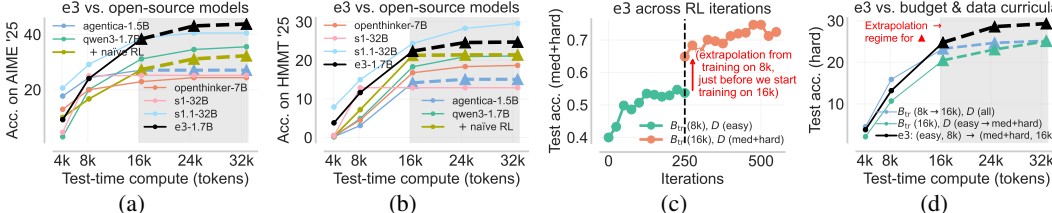

Figure 8: ***RL training with coupled curricula.*** In the above figure, the shaded area indicates the extrapolation regime. **(a), (b):** `e3` achieves *state-of-the-art* performance across models ¡ 2B. **(c):** extrapolation gain from switching to a longer budget during training **(d):** coupled curriculum outperforms data and budget curricula

***Why does this happen?*** Given a dataset, training on budgets smaller than the length of a typical response for the base model penalizes in-context exploration early in training. This results in overly short solutions (see Fig. 7(d)) that are mostly exploitative. When projected to our $p^k$ model from Sec. 4, this means that at overly short budgets, RL mainly attempts to improve the failure probability $p$ of the best guess response, and does not learn to increase $k$ which corresponds to chaining asymmetries. To increase $k$, it needs to be able to learn to increase the number of attempts and requires a large enough budget. But the budget cannot be too large either.

***How can we avoid challenges with training on a fixed dataset and length budget?*** One approach to avoid the above challenges is to incorporate a curriculum that varies $B_{\mathrm{tr}}$ over training. However, this alone is insufficient because, as shown above, training on hard problems with short budgets suppresses in-context exploration. On the other hand, we can design a curriculum over the difficulty level and set $B_{\mathrm{tr}}$ to a high enough value. However, this presents optimization challenges (see App. I for a detailed study of this on CDOWN). **In a nutshell**, a curriculum that only varies the training budget or the dataset composition is insufficient to incentivize in-context exploration. To mitigate this problem, we propose a "coupled" curriculum over data composition and training budget.

## 6.1 OUR RECIPE e3: COUPLED CURRICULUM FOR IN-CONTEXT EXPLORATION

We develop a ***coupled curriculum*** that varies the training budget $B_{\mathrm{tr}}$ and problem difficulty in a coordinated fashion during RL on a base model with asymmetries. We refer to our recipe (chained asymmetries, negative gradient, and the coupled curriculum) as `e3`: *exploration enables extrapolation*.

***Key insight for curriculum design.*** We simplify curriculum design by fixing the dataset at each stage and progressively increasing task difficulty in a stage-wise manner, from easy to hard. Now, at each curriculum stage $i$, we define a dataset $D_i$ and focus on selecting an appropriate token budget $B_{\mathrm{tr},i}$. The goal is to choose $B_{\mathrm{tr},i}$ such that training with this budget encourages *in-context exploration* when training on $D_i$. That is, RL should reward successful reasoning traces that successfully chain asymmetries, within budget $B_{\mathrm{tr},i}$. This ensures that the resulting policy can extrapolate to longer sequences and provides a strong initialization for the next stage $i+1$, where the token budget increases to $B_{\mathrm{tr},i+1}$. At the same time, for optimization to be efficient, the budget $B_{\mathrm{tr},i}$ should be as small as possible while still accommodating most valid completions from $\pi_i$. Balancing these desiderata, we formalize the choice of $B_{\mathrm{tr},i}$ via the following optimization as a thumb rule:

$$B_{\mathrm{tr},i}^{\star}(D_i) = \underset{B \geq B_0}{\arg\min} B \quad \text{s.t.} \ J(\pi_i; D_i, 2 \cdot B) \ \leq \ \kappa \cdot J(\pi_i; D_i, B), \ \ \kappa > 1 \tag{2}$$

where $J(\pi; D, B)$ denotes the performance of $\pi$ at budget $B$ on dataset $D$, and the budget $B_0$ denotes a reasonable minimal length for $\pi$ on dataset $D_i$, *e.g.*, $B_0$ can be the average token length of responses from $\pi$ on $D_i$. **In practice,** we solve the optimization over $B$ by restricting to a fixed set of training budgets: 4k, 8k, 16k. We find the above strategy of choosing the token budget to be a useful heuristic for greedily choosing the budget $B_{\mathrm{tr},i}$ at stage $i$ of the curriculum in a way that incentivizes in-context exploration. *E.g.*, setting $\kappa = 1.2$, we find 8k to be the best choice for training on easy problems (observe that the trained model satisfies the condition in Eq. 2 at $\kappa = 1.2$ in Fig. 7(a)). Following this, `e3` fine-tunes the Qwen3-1.7B base model on easy problems in DMATH at $B_{\mathrm{tr}}$ of 8k, and subsequently continues training on medium and hard problems in DMATH with a token budget of 16k. For training on medium/hard problems in DMATH, we can also optimize the training budget, as we did for the run on easy problems. From Fig. 7(a), we note that the model trained with a token budget of 8k extrapolates compute to a budget of 16k and even 24k on AIME '25, after which the gains start diminishing. We find similar extrapolation performance on medium and hard problems in DMATH. Thus, we can safely train on a budget of 16k or 24k on this set, and due to GPU memory

| Model | AIME 2025 | | | | | | HMMT 2025 | | | | | |
|---|---|---|---|---|---|---|---|---|---|---|---|---|
| | $k$=1 | 2 | 4 | 8 | 16 | 32 | $k$=1 | 2 | 4 | 8 | 16 | 32 |
| Qwen3-1.7B | 35.5 | 41.4 | 47.0 | 52.4 | 58.3 | 65.2 | 22.2 | 27.3 | 33.0 | 39.5 | 46.7 | 54.9 |
| R1-distill-Qwen-1.5B | 23.1 | 29.2 | 34.5 | 40.1 | 46.3 | 52.5 | 12.5 | 19.1 | 24.3 | 27.9 | 36.1 | 42.8 |
| Nemotron-Reasoning-1.5B | 33.6 | 38.5 | 43.6 | 48.9 | 53.8 | 58.0 | 17.4 | 22.5 | 29.6 | 35.2 | 40.7 | 45.0 |
| **e3-1.7B (Ours)** | **43.8** | **51.1** | **56.7** | **60.8** | **64.0** | **67.2** | **24.7** | **30.4** | **37.0** | **44.1** | **50.8** | **56.1** |

Table 1: *Final results with `e3` on AIME/HMMT'25*: We measure `pass@k` (%) on AIME'25 and HMMT'25 for our 1.7B model obtained by post-training the Qwen3-1.7B base model on DMATH with our recipe `e3`, comparing with strong <2B reasoning models. Following Sec. 6.1, we use a coupled task and budget curriculum during ((i) train on easy problems at $B_{tr}$=8k, and then (ii) on medium+hard ones at $B_{tr}$=16k). Note that unlike recent trends (Yue et al., 2025) that show RL training improving pass@1 at the cost of pass@k for a higher $k$, we note that `e3` trained models improve performance by not just sharpening the base model distribution around high reward traces, but by actually chaining asymmetries and discovering new solutions with longer traces.

constraints, we chose to train on the shorter of the two (16k). Finally, in Fig. 8(c), we show that the model produced by `e3` by training on easy problems at the end of the first stage does extrapolate well, which is helpful to kickstart RL training when we move from the budget of 8k to 16k. Concretely, we observe a ≥10% performance gain with extrapolation. We also show that our coupled curricula outperforms separate curriculums that either only vary the token budget ($B_{tr}$) or the data mixture ($D$) in Fig. 8(d). Finally, we illustrate the efficacy of the above curricula design on CDOWN in App. H.

## 6.2 FINAL RESULTS WITH E3: A *State-of-the-art* <2B MODEL ON AIME/HMMT'25

***Extrapolation to 32k with `e3` ≫ s1.*** In Fig. 8(a,b), we compare the performance of a Qwen3-1.7B model fine-tuned using `e3` with open-source models, including some 7B and 32B models. As shown, at a test-time token budget of 32k tokens, `e3` achieves state-of-the-art performance on AIME'25 and HMMT'25, within a model class of size <2B. We outperform the best model in this class by >8% on AIME'25 in terms of peak performance, and show that our model, trained only up to a budget of 16k, extrapolates better than other models including s1.1-32B (Muennighoff et al., 2025) and OpenThinker-7B (Team, 2025) when we extrapolate them to 32k output tokens. In principle, one can simply force the model (trained even with SFT) to use more test-time compute by intervening its output trace with an appended prompt (*e.g.*, by appending "Wait" to an output trace as suggested in

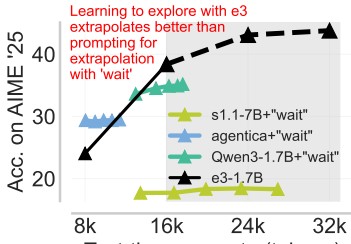

Figure 9: `e3` (w/o "wait") is superior when extrapolating to larger budgets, compared to budget forcing with "wait" prompt 2/4/6/8 times.

s1 (Muennighoff et al., 2025)). Interestingly, Fig. 9 shows that compared to budget forcing via "Wait", `e3` achieves substantially better scaling, without any form of prompting or budget forcing.

***Improving `pass@32` with `e3`.*** In Tab. 1, we also report the `pass@k` performance, comparing `e3` with other models of a similar size. We find that our final model at the end of second stage of training on a budget of 16k outperforms other models on higher values of $k$, on AIME and HMMT '25. We especially note the comparison against the Nemotron-Reasoning-1.5B model (Liu et al., 2025a) trained with a prolonged RL training recipe on a broader dataset, including our training data. This model consistently improves `pass@16` performance during RL training (Liu et al., 2025a). To concretely describe our estimation procedure, we used 128 rollouts per prompt to compute a bootstrapped estimate (Chen et al., 2021) of the `pass@k` performance for $k = 1, 2, \ldots, 32$, beyond which it would require substantially many more rollouts to get a reasonable high confidence pass@k estimate. We conclude that `e3` is able to improve over pass@32 of the base model, meaning that it does go beyond distilling the pass@32 policy corresponding to the base model into a better policy.

> **Takeaways: Coupled data & budget curriculum structures exploration during training.**
>
> - RL with fixed $B_{tr}$, $D$ hurts extrapolation: (i) short $B_{tr}$ curtails chaining on hard DMATH as budget is overrun; (ii) large $B_{tr}$ leads to over-exploratory behaviors on easy ones.
> - We propose a coupled curriculum `e3`: at each stage, given $D$, choose smallest $B_{tr}$ such that chaining more asymmetries till a budget of $2 \cdot B_{tr}$ is positively rewarded at RL initialization.

# 7 DISCUSSION AND CONCLUSION

We show that in-context exploration is a core capability to enable extrapolation of test-time compute in LLMs. Therefore we build a recipe that amplifies in-context exploration. Our recipe e3, leverages (1) asymmetries in the base model, (2) negative gradients during RL training, and (3) a coupled curriculum over data and token budget to train a model that can perform in-context exploration. Applied to the Qwen3-1.7B model, our method achieves state-of-the-art performance on the AIME/HMMT'25 benchmarks, with particularly strong gains in the extrapolation regime. We also show e3 improves pass@k over the course of training, for values of $k$ upto 32 that we evaluate. There are multiple implications of our work and interesting directions that future work can build upon.

**Sharpening vs in-context exploration.** A number of concurrent results either directly (Yue et al., 2025) or indirectly (Shafayat et al., 2025; Zhao et al., 2025b; Shao et al., 2025; Prabhudesai et al., 2025) argue that RL training on LLMs sharpens the base model's distribution. In contrast to this, our study shows that if we can utilize a coupled curriculum on top of a base model that admits asymmetries, RL can actually enable chaining new asymmetries, resulting in an increase in length, indicating the presence of structured exploration. This behavior is distinct from traditional sharpening that corresponds to cloning one (or few) of the responses sampled from the base model. In fact, our conceptual study in the $p^k$ model in Section A also highlights these two distinct phases during RL: an initial in-context exploration phase where negative gradients lead to an increase in response length and the policy learns to utilize test-time compute for better exploration, followed by a phase where it sharpens to the best traces found thus far. The design of e3 enables it to operate in the former phase. We believe concurrent works that finds RL largely sharpens the model operate in the second regime by training on data that does not require chaining asymmetries or operating with a very low training budget such that chaining is impossible. As a result, models trained purely in the sharpening regime may behave similarly to the base model with an alternate prompt, with RL perhaps offering little more than an implicit prompt tuning effect. But we would not expect this for the chaining regime.

**Connection with dense progress rewards.** While e3 utilizes a coupled curriculum, this curriculum is closely connected with the use of dense rewards, as prescribed by our prior work (Qu et al., 2025b; Setlur et al., 2024). To see why, note that one can reparameterize coupled curriculum into a single round of training with dense rewards applied to short segments of the output response, perhaps in a similar way as Qu et al. (2025b); Qi et al. (2025). Therefore, the success of the coupled curriculum approach in e3 at improving performance and not only in reducing total training compute perhaps hints at future success with dense rewards at scale, with initial results showing that dense rewards help larger models already being shown in the community (Wang et al., 2025a). We encourage readers to explore the connection between curriculum and dense rewards further.

**Introducing new asymmetries.** The conceptual model behind e3 applies with any asymmetry, though most experiments in this paper utilize only the verification-generation gap. It would be interesting to identify other asymmetries and study methods to imbue base models with these asymmetries. Definition 4.1 in Section 4 provides a starting point to define these asymmetries.

**Is curriculum fundamentally needed?** A natural question is whether curriculum is fundamentally necessary as we vary model sizes and capabilities. Unlike supervised learning on a fixed dataset, online RL generates its own rollouts. Reinforcing chaining behavior via negative gradients (Sec. 5) requires that such chaining reliably improves performance on training problems much more substantially compared to sampling diverse traces that do not chain asymmetries. This likely necessitates specific training configurations regardless of model size with standard outcome-reward RL, or the use of dense rewards (as discussed above). While larger models may admit simpler curricula, deliberately using currciulum or dense rewards as inspiration may be critical.

**Explicit exploration bonuses.** In our runs, the main issue hindering us from benefits of further scaling of output length during RL is the repetition bias in the base model, where it tends to repeat previously-generated segments in its trace beyond a certain output length. This repetition bias inhibits the efficacy of in-context exploration beyond a certain output length and as a result inhibits further test-time scaling. We believe that explicit exploration bonuses that enable the model to search for tokens in this regime would result in even better in-context exploration.

Finally, our study is limited in terms of model scale and domain. Future work should explore how e3 generalizes to larger model scales and other reasoning domains.

ACKNOWLEDGEMENTS

We thank Christina Baek, Yuxiao Qu, Anikait Singh, Yoonho Lee, Max Sobol Mark, Zheyuan Hu, Seohong Park, Bhavya Agrawalla, Sang Michael Xie, Paria Rashidinejad, and the rest of the AIRe lab at CMU for informative discussions, feedback, input on our results, and a previous version of this paper. We thank Yuxiao Qu for help with debugging implementations and infrastructure. The main large-scale experiments in this paper utilized H100 GPU resources from the Orchard cluster in the FLAME center at CMU for which we especially thank Graham Neubig and Chenyan Xiong for their generous support, and TPUs from Google Cloud. We thank Oumi for providing us with resources that supported the experiments on the Countdown domain. This project is supported by funding from the Office of Naval Research under N00014-24-1-2206 and a Schmidt Sciences AI2050 Fellowship. AS is supported by JP Morgan PhD fellowship. This paper does not reflect the opinions of employers or other parties.

## 8 REPRODUCIBILITY STATEMENT

We have taken multiple steps to ensure the reproducibility of our work. A detailed description of the training setup, including datasets (DMATH, CDOWN, MULT, and benchmarks such as AIME'25 and HMMT'25), model architectures, and hyperparameters, is provided in the main text (Sections 3–6) and in the appendices (Appendix E–I). The theoretical results and proofs underpinning our claims about asymmetries and negative gradients are included in App. A and App. F. To aid experimental reproducibility, we describe our evaluation protocols (e.g., pass@k computation, bootstrapped confidence intervals) in Sec. 6.2 and App. N. We also provide ablation studies and additional results across different training budgets, curricula, and datasets in Appendix G–K. An anonymous link to source code and scripts for data preprocessing, training, and evaluation is included in the supplementary materials. Together, these resources allow independent researchers to replicate both the theoretical and empirical findings presented in this paper.

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

# Appendices

## A  ANALYZING NEGATIVE GRADIENT DYNAMICS IN THE $p^k$ MODEL

In this section, we introduce a didactic $p^k$ model, where an LLM samples $k$ independent actions sequentially, verifies them (with a perfect accuracy), and terminates immediately after the correct one is produced. In this section, we introduce a didactic setup where verification is perfect (and hence, there is a high VG gap), and formalize the intuitions regarding negative gradient from the previous section.

**Didactic analysis setup.** We consider a Markov decision process (MDP) (Puterman, 1994) with action space $\bar{\mathcal{A}} = \mathcal{A} \cup \{\texttt{stop}\}$, where $\mathcal{A} = [100]$ are standard actions and $\texttt{stop}$ is an early "stopping" action (like $\texttt{EOS}$) that terminates the trace. For simplicity, we consider policies parametrized as a softmax bigram model $\pi_M(a_{t+1} \mid a_t)$: in this model, the policy only retains one token in its history and is parameterized by a softmax over logits described by bi-grams, i.e., $\pi_M(a_{t+1}|a_t) \propto \exp(M(a_t, a_{t+1}))$. In this bi-gram model, the current state $s_t$ always matches the previous action $a_{t-1}$, and $a^\star \in \mathcal{A}$ denotes the optimal action. In a rollout $a_1, ..., a_t$, the initial action $a_1$ is sampled from a fixed $\pi_0$. For $t > 1$, a learner policy samples an action $a_t \sim \pi(\cdot|a_{1:t-1}) \in \triangle(\mathcal{A})$. The MDP terminates with reward 1 at time $t$ if $a_t = a^\star$, and with reward 0 if $a_t = \texttt{stop}$ (stops too early), or $t > B_{\text{tr}}$ (budget is exhausted before a correct response). The policy is initialized to one that puts a high probability mass on choosing $a = \texttt{stop}$. Details are in App. G.

We say that the model **learns to explore in-context** if it learns to never play $\texttt{stop}$ for any $t$ (no early stopping), until $a^\star$ is observed, i.e., increasing $k$ in $p^k$. On the other hand, **classical exploration** amounts to upweighting $\pi(a^\star \mid a_{1:t-1})$ without reducing $p(\texttt{stop})$, i.e., improving $p$ in $p^k$.

*Finding 1: Negative gradient increases length until $p(a^\star)$ is reasonably high.* In Fig. 10(a), standard GRPO ($B_{\text{tr}} = 100$) increases average response length from 15 to 45 at budget, driven by the drop in the marginal probability of stopping early $p(\texttt{stop})$ (Fig. 10(c)). After multiple RL iterations with negative gradients, the average number of attempts per trace is sufficiently large, and the learner can sample $a^\star$ with non-trivial probability in any given trace. Once this happens (Fig. 10(c)), in our simple bigram setup, the model rapidly upweights the likelihood of one-step transitions to $a^\star$, resulting in a phase transition where reward increases as length drops. In contrast, GRPOMask (Fig 10(b)) fails to improve reward or increase length. The first phase is akin to chaining more asymmetries in LLMs and results in a longer response length. In our LLM benchmarks, however, we do not see the same phase transition since finding "shortcuts" to correct responses is considerably more difficult. Moreover, the LLM is conditioned on an entire history and learns to utilize the history carefully in the first phase. This makes it unlikely for it to quickly learn to reduce length substantially even if it transitions into this second phase on some problems.

*Finding 2: Negative gradient improves coverage by increasing entropy of $\pi_M(\cdot \mid a_{1:t-1})$.* When $\pi_M$ samples a highly likely yet incorrect action, the negative gradient computed on this sample increases entropy by moving probability mass onto less-seen modes of the distribution, including $a^\star$. Note that no explicit entropy bonus is applied. We show this formally in Theorem A.1 where

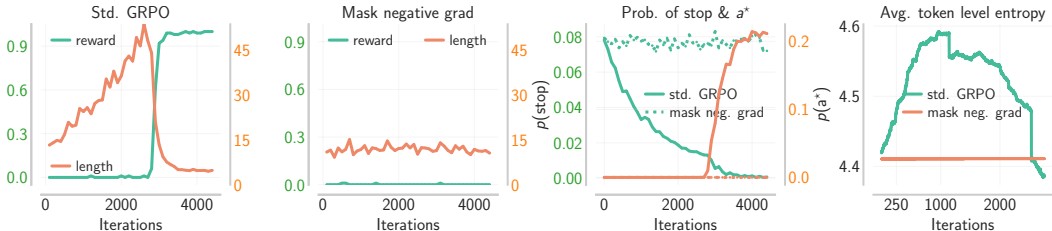

Figure 10: *Negative gradients in the $p^k$-model.* Negative gradients push down $p(\texttt{stop})$ during training (c), increasing length (a) and entropy of the next action distribution (d) to accommodate more in-context exploration, only decreasing them when $a^\star$ is discovered. In contrast, positive gradients rarely change $p(\texttt{stop})$ or entropy.

we prove that upon sampling a highly likely incorrect action with probability $p$, GRPO update with a negative gradient results in an entropy increase of $\approx p^2$ when all other actions, including $a^\star$ are highly unlikely. We note this empirically as well in Fig. 10(d), where conditional entropy increases across states, until $a^\star$ is discovered, after which it drops sharply as the positive gradient rapidly moves mass onto $a^\star$ within a few iterations.

**Theorem A.1** (Negative gradient increases entropy when $a^\star$ is unlikely; formal version in Thm. J.3)*.*
*At state* $\mathbf{s}$*, if the most likely action under $\pi$ is $a_1 =: \arg\max_{a'} \pi(a'|\mathbf{s}) \neq a^\star$, then, for any $\pi$, a negative stochastic gradient step increases the entropy of $\pi(\cdot|\mathbf{s})$ with prob. $\geq \pi(a_1|\mathbf{s})$. Additionally, in a suitable regime of $\pi$, the increase $\gtrsim (\pi(a_1|\mathbf{s}) - \pi(a_2|\mathbf{s}))^2$, where $a_2$ is second most likely after $a_1$. In contrast, in the absence of the negative gradient, the entropy is preserved with prob. $1 - \pi(a^\star|\mathbf{s})$.*

**Connecting back to our findings in Sec. 5.** From the above analysis in the bi-gram model it is clear that the recovered from the `<EOS>` token indeed moves to the optimal action under certain conditions. But, more generally, for our experiments on LLMs in Sec. 5, all we can say is that the negative gradient will decrease the probability of `<EOS>`. So, it is unclear why the length increases and doesn't drop from the negative reinforcement. One explanation for the length increase in Fig. 5 is that when we train LLMs with RL we fine-tune a base model and don't train from scratch. So, the priors in the base model have a huge effect on where the recovered mass from the `<EOS>` token lands. If the base model has a prior to chain asymmetries (verification-generation (VG) chains), then reducing mass from the `<EOS>` token increases mass on other likely actions in the base model, including the action of "chaining more". If the base model does not present this prior, *e.g*, if the base model does not present a verification-generation gap on a task, then we would not expect to see a length increase. For this, please see our experiments on MULT (base model presents no VG gap on this task) vs. MULT-V(base model presents VG gap) in Fig 5. Here, it is clear that the length increase during RL training is more prominent in the task where the VG gap is present. This shows that the base model prior has a huge role to play in where the probability mass from the `<EOS>` token goes.

## A.1 ADDITIONAL DISCUSSION ON THE DIDACTIC ANALYSIS

First, we comment on exploration and meta-exploration in RL, and how negative gradients in our didactic setting can connnect one to the other in the presence of asymmetries. Next, we introduce some details for the policy parameterization and training.

**Negative gradients boost exploration, which in the presence of asymmetries incentivizes in-context exploration.** In Sec. 5 we showed how negative gradients can boost exploration in RL, and in the presence of asymmetries in the base model, lead to more chained asymmetries and longer responses – a phenomenon we call in-context exploration. Here, we present a theoretical result that explains why negative gradient can incentivize the more "traditional exploration" in RL, in our didactic bi-gram model. Since verification is perfect in our bi-gram model, any policy in our policy class always stops at the `stop` token. Thus, an increase in exploration leads to longer traces, and more chained asymmetries. As a result, in this setting, we can view an improvement in exploration as an improvement in meta-exploration (or in-context exploration), driven by negative gradients.

**Parameterization of the policy class.** We parameterize the policy class as a softmax policy, where the probability of next action $a_{t+1}$, at state current $a_t$ (in a bi-gram model current state is equivalent

to the previous action) is parameterized with the vector of logits $[M(a \mid a_t)]_{a \in \bar{\mathcal{A}}}$, i.e.:

$$\pi_M(a_{t+1} \mid a_t) = \frac{e^{M(a_{t+1} \mid a_t)}}{\sum_{a' \in \bar{\mathcal{A}}} e^{M(a' \mid a_t)}}, \quad a_{t+1} \in \bar{\mathcal{A}}, a \in \mathcal{A} \tag{3}$$

where $M = [M(a^+ \mid a)]_{a^+ \in \bar{\mathcal{A}}, A \in \mathcal{A}}$ can be expressed as a matrix in $\mathbb{R}^{(K+1) \times K}$. Note that the cuurent state can never be the $\texttt{stop}$ action, since a $\texttt{stop}$ always terminates the MDP.

**Training details.** We set the initial distribution $\pi_0$ to be the uniform distribution over all actions except $a^\star$, i.e., $\pi_0(a^\star) = 0$. For each state $\mathbf{s}$, the policy is first initialized with random values of $M(\cdot \mid \mathbf{s})$ in $[-3.0, 3.0]$, and then we set $M(\texttt{stop} \mid \mathbf{s}) = 4.0$, $M(a^\star \mid \mathbf{s}) = -4.0$, which mimics the setting where the probability of sampling the stop action is higher than any random action, and the probability of sampling $a^\star$ is lower than any random action. We train with a learning rate of $1e\text{-}2$ and use stochastic gradient descent to update the policy where a single update samples a random trajectory $\tau$, starting from a random state sampled from the initial state distribution $\pi_0$, by running the policy until termination of the MDP. We then compute the policy gradient term, by averaging the policy gradient loss over the tokens in the trajectory $\tau$: $1/|\tau| \cdot \sum_{i \in |\tau|} \log \pi_M(a_i \mid a_{1:i-1}) \cdot A(a_i, a_{1:i-1})$.

# B    TESTING EXTRAPOLATION OF OPEN-SOURCE MODELS

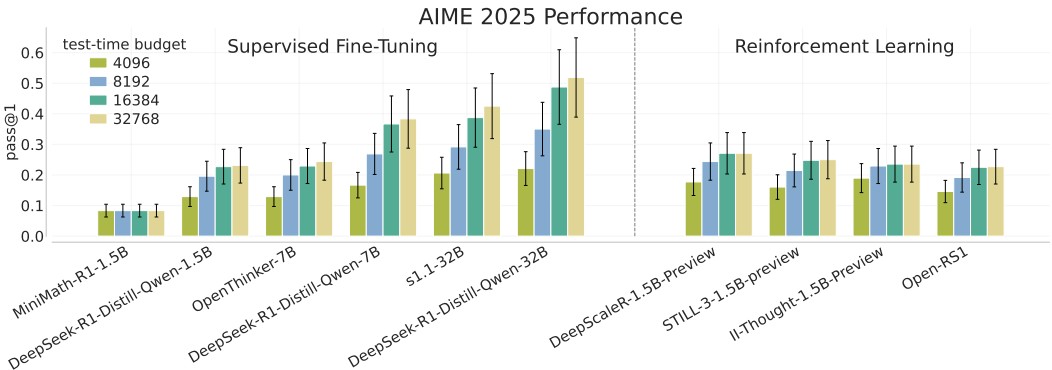

Figure 11: *Extrapolation of test-time compute:* We plot the performance (pass@1) on AIME 2025 at different test-time compute budgets across multiple open-source models of different sizes, trained with SFT or RL.

**Extrapolation on AIME 2025.** Extrapolation (i.e. the chaining of generation, verification, refinement, etc.) can potentially extend LLM performance after training, and do so beyond the context length the model was originally trained on. To evaluate this properly, we need sufficiently challenging problems that allow meaningful expressiveness in reasoning beyond small context lengths. The math problems associated with AIME align with this, and our evaluations prioritize AIME 2025 to attempt to mitigate any potential data contamination in the models' training sets from previous years of AIME. The goal of the experiment is to measure the extent to which test-time compute influences overall model performance as context length increases, with the expectation that increasing output length allows models to "reason" for longer periods, continuing the extrapolation process, and ultimately arriving at the correct answer more frequently.

**Experiment setup.** Inference for every open-source model was performed using Oumi through data-parallel SGLang. All models had inference run with a max output length of approximately 32k tokens, though some are slightly lower due to this exceeding their max context length when combined with the prompt. The exact inference hyperparameters are described in Table 2. After inference, the model responses were truncated from the right side until the number of remaining tokens present was equal to the specified test-time budget. 16 responses were collected for every problem in AIME with the specified inference settings, and the Pass@1 rate was calculated by averaging over these 16 responses. Final answers were extracted using a regular expression for the boxed portion of the answer, with correct answers marked as passing and incorrect or incorrectly parsed answers marked as nonpassing. The prompt used is in Box B.1, and the problems were taken from the FVU AIME 2025 dataset on HuggingFace[1].

---

[1] https://huggingface.co/datasets/FVU/AIME_2025

> **Box B.1: AIME Evaluation Prompt Template**
>
> You will be given a math problem. Solve the problem step by step. Output your final answer in the form of \\boxed{your answer}. Problem: {problem}

| Model | Temp. | Top $p$ | Rollouts | Max New Tokens | Model Max Length |
|---|---|---|---|---|---|
| MiniMath R1-1.5B | 0.6 | 0.95 | 16 | 32768 | 40960 |
| DeepSeek R1-Distill-Qwen-1.5B | 0.6 | 0.95 | 16 | 32768 | 40960 |
| OpenThinker-7B | 0.6 | 0.95 | 16 | 31000 | 32768 |
| DeepSeek-R1-Distill-Qwen-7B | 0.6 | 0.95 | 16 | 32768 | 40960 |
| s1.1-32B | 0.6 | 0.95 | 16 | 31000 | 32768 |
| DeepSeek-R1-Distill-Qwen-32B | 0.6 | 0.95 | 16 | 32768 | 40960 |
| DeepScaleR-1.5B-Preview | 0.6 | 0.95 | 16 | 32768 | 40960 |
| STILL-3-1.5B-preview | 0.6 | 0.95 | 16 | 32768 | 40960 |
| II-Thought-1.5B-Preview | 0.6 | 0.95 | 16 | 32768 | 40960 |
| Open-RS1 | 0.6 | 0.95 | 16 | 32768 | 40960 |

Table 2: Inference parameters used for generating the extrapolation plots in Figure 2.

**Results.** The results in Figure 11 show that as the maximum number of output tokens increases, every model capable of "reasoning" is able to attain a higher Pass@1 rate, with performance generally saturating at 16k tokens with relatively minor improvements at 32k. We do not observe this with MiniMath-R1-1.5B, and we suspect this is due to its fine-tuning focusing solely on smaller math problems trained with supervised fine-tuning, likely resulting in catastrophic forgetting of the ability to continuously extrapolate. Interestingly, we do not see a strong improvement in extrapolation behavior among models tuned with reinforcement learning compared to DeepSeek R1-Distill-Qwen-1.5B, which was trained with supervised fine-tuning. We suspect that this is likely due to the nature of the distillation data from the R1 model, which, if varied sufficiently in length, could avoid the length bias normally learned from supervised fine-tuning, while still teaching the model to perform extrapolation.

In addition to our evaluation on AIME2025, we also evaluate extrapolation on the older math benchmark of MATH-500. We show the extrapolation performance of open source models below. We also compare the performance with our `e3`-1.7B model.

| Model / Test-time compute (tokens) | 4096 | 8192 | 16384 | 24576 | 32768 |
|---|---|---|---|---|---|
| agentica-1.5B | 74.65 | 83.18 | 84.25 | 84.25 | 84.25 |
| OpenThinker3-1.5B | 47.90 | 71.18 | 82.58 | 85.85 | 86.53 |
| Qwen3-1.7B | 55.98 | 75.28 | 83.53 | 85.10 | 85.45 |
| OpenThinker-7B | 63.48 | 73.35 | 78.93 | 79.93 | 80.10 |
| s1.1-32B | 73.40 | 83.43 | 86.98 | 87.93 | 88.23 |
| **e3-1.7B (Ours)** | **66.53** | **81.38** | **86.38** | **86.88** | **86.95** |

Table 3: **MATH500 evaluation.** Accuracy (%) across varying test-time compute budgets. Our **e3-1.7B** consistently outperforms prior sub-2B models, surpasses the 7B SFT-trained **OpenThinker-7B**, and approaches the performance of **s1.1-32B** (over 18× larger) at longer token budgets. All models are evaluated with top_p = 0.95, temperature = 0.6, and accuracy averaged across 8 rollouts per prompt.

## C EXPANDED DISCUSSION OF RELATED WORK

**Scaling test-time compute via long CoT reasoning.** Prior work explores a number of avenues for scaling test-time compute, including majority voting (Wang et al., 2022), best-of-n sampling, and beam search (Setlur et al., 2024; Snell et al., 2024), as well as sequential self-correction (Qu et al., 2024; Kumar et al., 2024). More recent results indicate that training models to use test-time compute to generate longer chains of thought (CoT) that combine verification, search, and self-correction – all

in a free-form manner, performs better (DeepSeek-AI et al., 2025; Team et al., 2025; OpenAI et al., 2024), resulting in widespread open-source reproduction efforts (Face, 2025; Yeo et al., 2025; Zeng et al., 2025b; Luo et al., 2025b). We situate our work in the paradigm of long CoT reasoning.

**Test-time extrapolation.** The true benefit of test-time scaling is consistently improving performance as we extrapolate test compute. While prior work tests the model's performance on budgets longer than the training budget Zeng et al. (2025a); Luo et al. (2025a), they do not explain the relationship between the training recipe and the extrapolation, like we aim to do in our work. Other works perform extrapolation by explicitly prompting models to generate more tokens when a response terminates Muennighoff et al. (2025); Aggarwal & Welleck (2025), whereas, we show that models that learn to explore in-context extrapolate test compute better than prompting-based approaches (Fig. 8). In particular, we study the role played by the base model, training algorithm (RL), as well as data mixtures and token budgets, on the ability to extrapolate. Furthermore, prior work Setlur et al. (2025b) has investigated scaling when train and test budgets are the same, but we expand the scope of this comparison substantially.

**Exploration in test-time scaling.** While prior works have shown the importance of the base model's ability to conduct exploration (Gandhi et al., 2025; Liu et al., 2025b), we discover that it is crucial for extrapolation. We show that the negative gradient in RL incentivizes chaining multiple asymmetries and leads to longer response length, and better performance. SFT alone does not provide this kind of chaining or exploration benefits. Our analysis is orthogonal to theoretical works Setlur et al. (2025b); Swamy et al. (2024), which shows that RL performs better than SFT, but from a statistical perspective, whereas our argument is more focused on the learning dynamics. Concurrent work builds techniques to boost exploration during RL via advantage normalization (Li et al., 2022; Yu et al., 2025) or PPO clipping (Yu et al., 2025), and these techniques can be combined with e3, but they do not study the role of negative gradients in learning to explore. Finally, Wang et al. (2025b) briefly remarks about the role of policy gradient loss and entropy when running RL with only a few examples. Our study investigates the underlying mechanism of negative gradients increasing length and entropy.

**Data and length curricula.** Recent works have also investigated using a curriculum on problem difficulty Team et al. (2025); Xie et al. (2025); Shi et al. (2025) and output length Luo et al. (2025b); Liu et al. (2024) during RL training. Their motivation stems primarily from an efficiency standpoint: avoiding zero advantage updates Shi et al. (2025); Yu et al. (2025), efficient optimization Luo et al. (2025b), or efficiency of using test-time compute Qu et al. (2025b). While we do make similar observations regarding each curriculum individually, perhaps our most interesting finding is that carefully coupling both data and budget curricula can lead to much better performance and extrapolation, beyond merely some gains in efficient training. We show that training on hard problems with short budgets often yields terse solutions that fail to extrapolate, while easy problems with long budgets can cause optimization issues or verbose outputs. Thus, curricula must be carefully designed to support effective extrapolation. Conceptually, our curricula are most related to dense progress rewards (Qu et al., 2025b; Setlur et al., 2024), in the sense that curricula incentivize different degrees of progress for different questions, at different points in training.

## D    PERSPECTIVES ON FUTURE WORK

There are a number of implications of our work and a number interesting directions that future work should build upon. We list the main technical implications and open questions below.

- **Sharpening vs in-context exploration.** A number of concurrent RL results either directly (Yue et al., 2025) or indirectly (Shafayat et al., 2025; Zhao et al., 2025b; Shao et al., 2025; Prabhudesai et al., 2025) argue that RL training on LLMs sharpens the base model's distribution, as also previously studied by Huang et al. (2024). In contrast to this, our study shows that if we can utilize a coupled curriculum on top of a base model that admits asymmetries, RL can actually enable chaining new asymmetries, resulting in an increase in length, indicating the presence of structured exploration. This behavior is distinct from traditional sharpening that corresponds to cloning one (or few) of the responses sampled from the base model. In fact, our conceptual study in the $p^k$ model in Section A also highlights these two distinct phases during RL: an initial in-context exploration phase where negative gradients lead to an increase in response length and the policy learns to utilize test-time compute for better exploration, followed by a phase where it sharpens to the best traces found thus far. The design of e3 enables it to operate in the former phase. We believe concurrent works that

finds RL largely sharpens the model operate in the second regime by training on data that does not require chaining asymmetries or operating with a very low training budget such that chaining is impossible. As a result, models trained purely in the sharpening regime may behave similarly to the base model with an alternate prompt, with RL perhaps offering little more than an implicit prompt tuning effect. But we would not expect this for the chaining regime. A detailed study on separating these regimes, and identifying all the factors that draw RL training into these regimes is an interesting direction for both theoretical and empirical research.

- **Connection with dense progress rewards.** While e3 utilizes a coupled curriculum, this curriculum is closely connected with the use of dense rewards, as prescribed by our prior work (Qu et al., 2025b; Setlur et al., 2024). To see why, note that one can reparameterize coupled curriculum into a single round of training with dense rewards applied to short segments of the output response, perhaps in a similar way as Qu et al. (2025b); Qi et al. (2025). Therefore, the success of the coupled curriculum approach in e3 at improving performance and not only in reducing total training compute perhaps hints at future success with dense rewards at scale, with initial results showing that dense rewards help larger models already being shown in the community (Wang et al., 2025a). We encourage readers to explore the connection between curriculum and dense rewards further.

- **Introducing new asymmetries.** The conceptual model behind e3 applies with any asymmetry, though most experiments in this paper utilize only the verification-generation gap. It would be interesting to identify other asymmetries and study methods to imbue base models with these asymmetries. Definition 4.1 in Section 4 provides a starting point to define these asymmetries.

- **Is curriculum fundamentally needed?** A natural question is whether curriculum is fundamentally necessary as we vary model sizes and capabilities. Unlike supervised learning on a fixed dataset, online RL generates its own rollouts. Reinforcing chaining behavior via negative gradients (Sec. 5) requires that such chaining reliably improves performance on training problems much more substantially compared to sampling diverse traces that do not chain asymmetries. This likely necessitates specific training configurations regardless of model size with standard outcome-reward RL, or the use of dense rewards (as discussed above). While larger models may admit simpler curricula, deliberately using currciulum or dense rewards as inspiration may be critical.

- **Explicit exploration bonuses.** In our runs, the main issue hindering us from benefits of further scaling of output length during RL is the repetition bias in the base model, where it tends to repeat previously-generated segments in its trace beyond a certain output length. This repetition bias inhibits the efficacy of in-context exploration beyond a certain output length and as a result inhibits further test-time scaling. We believe that explicit exploration bonuses that enable the model to search for tokens in this regime would result in even better in-context exploration.

# E  ADDITIONAL EXPERIMENTS AND DETAILS FOR SECTION 4 (CHAINED ASYMMETRIES)

## E.1  DETAILS ON MULT AND MULT-V

**Data collection.** Both MULT and MULT-V consist of multiplication traces for solving a 5-digit $\times$ 5-digit multiplication problem. For the MULT task, we use a Llama3.2-3B instruction tuned model where the number of intermediate verification attempts is much lower in a trace when asked to solve a multiplication problem. In fact, it is not hard to see that, in general, for multiplication, generation of a trace may be as hard as verifying a generated one, as the only way to verify the entire trace is to re-attempt the multiplication or carry out a division with the computed target. We contrast this task with the MULT-V task, where the Llama3.2-3B models are first finetuned on traces from Qwen-32B-R1-Distilled and GPT-4o models. These traces contain multiple verification attempts that verify intermediate steps solving smaller multiplication problems, and the steps are part of an entire trace that attempts to solve the main multiplication problem involving two 5-digit numbers. For collecting data we used the prompt in Box E.1. In App. N Example 2, we also provide an example multiplication trace with verification attempts sampled by the base model in MULT-V. As we will see in Fig. 16, the absence of asymmetries in MULT leads to lower accuracy and verifications when compared to MULT-V, where asymmetries are present.

> **Box E.1: Prompt for generating MULT-V data**
>
> Multiply {num1} and {num2}. Please reason step by step, and put your final answer within \\boxed{}. At each step, try to verify your response if possible and prefix the line with "Check:". ¡think¿

| Hyperparameter | Values |
|---|:---:|
| train_batch_size | 256 |
| ppo_mini_batch_size | 64 |
| learning_rate | 5.0e-6 |
| kl_loss_coef | 0.001 |
| entropy_coeff | 0.001 |
| temperature | 1.0 |
| rollout.n | 16 |
| ppo_lowerclip_threshold | 0.2 |
| ppo_higherclip_threshold | 0.2 |

Table 4: Verl Sheng et al. (2024) hyperparameters used for MULT and MULT-V.

**Training details.** Hyperparameters for our experiments on MULT and MULT-V are given in Table 4.

### E.2 DETAILS ON CDOWN

**Training details.** Hyperparameters in CDOWN experiments follow the table below unless otherwise specified. In all of our CDOWN experiments, we take the fine-tuned Llama3.2-3B base model from Gandhi et al. (2025). For Fig. 4, we trained with $B_{\mathrm{tr}} = 512, 1024, 2048$ on problems with $3, 4, 5, 6$ candidates. The total number of datapoints we used was 40000, which were evenly split across the four difficulties.

| Hyperparameter | Values |
|---|:---:|
| train_batch_size | 128 |
| ppo_mini_batch_size | 32 |
| learning_rate | 1.0e-6 |
| kl_loss_coef | 0.001 |
| entropy_coeff | 0 |
| temperature | 0.6 |
| rollout.n | 8 |
| ppo_lowerclip_threshold | 0.2 |
| ppo_higherclip_threshold | 0.2 |

Table 5: Verl Sheng et al. (2024) hyperparameters used for CDOWN.

**Evolution of chained asymmetries at test time.** In Fig. 12, we show that as training progresses, responses with more chained asymmetries enjoy a greater improvement. If we move across any diagonal parallel to the main diagonal from top left to bottom right, we move across a constant attempt budget (*e.g.,* moving from 16 chained asymmetries × 1 pass to 8 chained asymmetries × 2 passes). Having sequential chained asymmetries become increasingly better than parallel rollouts as training progresses, indicating the exploitation of asymmetries in RL training. See example of chained asymmetry in App. N, Example 1.

### E.3 IN THE PRESENCE OF ASYMMETRIES, KL DIVERGENCE WITH BASE LLM REDUCES AS TRAINING TOKEN BUDGET INCREASES

In Fig. 13, we interestingly observe that training with higher $B_{\mathrm{tr}}$ results in a smaller token KL-divergence from $\pi_b$ all throughout training on countdown. On multiplication in the absence of

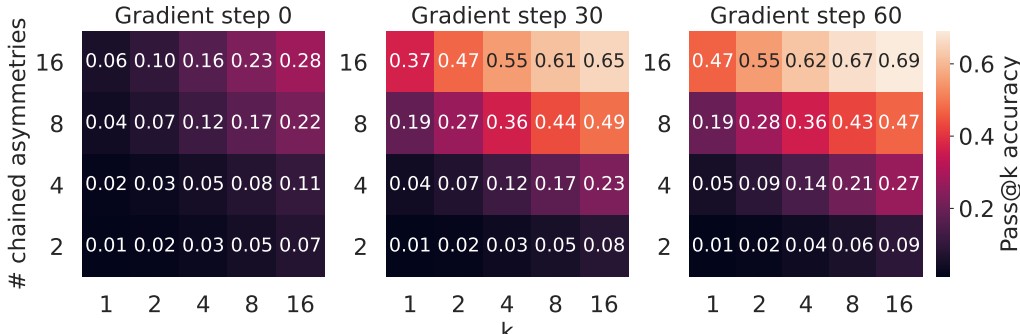

Figure 12: *Evolution of asymmetries during training on* CDOWN: More chained asymmetries lead to a greater improvement in pass@k performance across gradient steps.

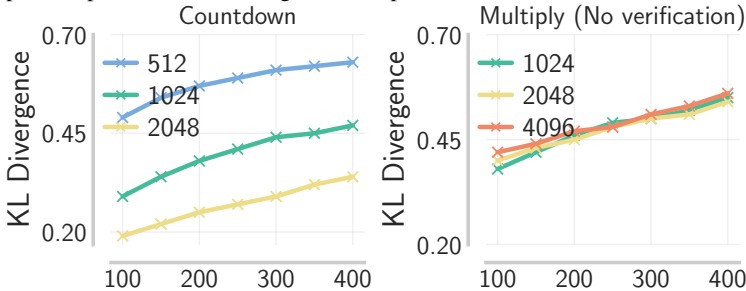

Figure 13: *KL-divergence with base LLM on* CDOWN *and* MULT: When running RL training on CDOWN and MULT with multiple training budgets (512, 1024, 2048 on CDOWN and 1024, 2048, 4096 on MULT) we note that the KL divergence is lower when running RL training on higher training budgets, when the base model presents asymmetries (in here the asymmetry is given by the verification generation gap on CDOWN).

asymmetries, the KL-divergence values are roughly similar for all $B_{\text{tr}}$. This means that when the verification-generation asymmetry is present, the training process deviates less from $\pi_b$ at each token, but is able to "chain" multiple verification and generation attempts together to improve accuracy, by learning to explore over the space of basic skills. Prior work argues that a model that deviates less from the base pre-trained model generalizes better on unseen prompts (Gao et al., 2019). If we were to apply this argument in our case, this means that models that are able to use asymmetries better should yield better performance on unseen prompts, especially with larger test budgets.

### E.4 VERIFICATION GENERATION GAP IN LARGER MODELS.

In Fig. 14(left), we empirically validate the verification-generation gap for a larger model (Qwen3-8B) on hard DMATH problems. As our $\text{detect}(q(p(\cdot), \tau))$ score increases (indicating traces with more verification-generation chains) the pass@$k$ performance consistently improves for $k \in \{1, 2, 4\}$. This monotonic trend shows that instances where the in-context verifier can more reliably distinguish good from bad trajectories are exactly those where more attempts afforded by an increased sampling budget translates into larger gains, confirming that the asymmetry captured by Definition 4.1 is not a small-model artifact but persists even at the 8B scale.

In Fig. 14(right), we empirically validate the verification-generation gap for one of the largest open source reasoning models (DeepSeek-R1, 685B parameters) on a set of 150 problems (8 rollouts per problem) in OpenMathReasoning Moshkov et al. (2025). We did not use DMATH for this since R1 solves all problems in the hardest set of DMATH with very high accuracy. Instead, we picked 150 random problems from OpenMathReasoning Moshkov et al. (2025) where the pass@1 rate (out of 32 generations) for Qwen2.5-Math-72B-Instruct run in TIR mode is less than 0.05. We find that on traces with more verification-generation chains ($\text{detect}(q(p(\cdot), \tau))$ in Definition 4.1) the pass@1 performance is higher than on traces with fewer chained asymmetries. This tells us that the asymmetry between verification and generation holds in even one of the largest (685B parameters) state-of-the-art reasoning models, as reflected in the model's reliance on exploiting this asymmetry to improve performance.

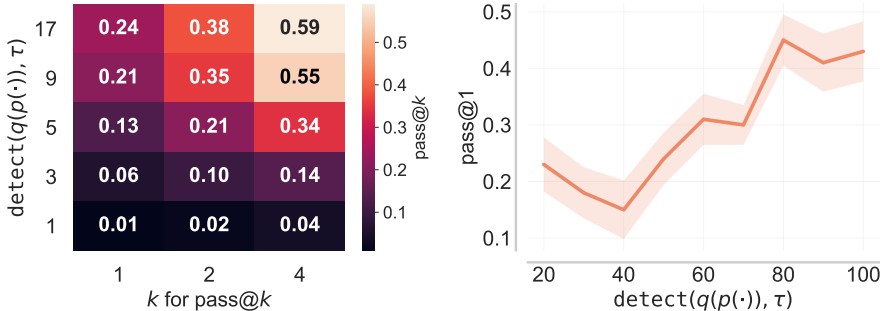

Figure 14: *Verification-generation gap in large models*: We compute our $\mathrm{detect}(q(p(\cdot), \tau))$ metric in Definition 4.1 on: *(left)* traces $\tau$ sampled from the Qwen3-8B model on hard problems in DMATH; and *(right)* traces $\tau$ sampled from the 685B parameter model DeepSeek-R1 (on a set of very hard problems in OpenMathReasoning (Moshkov et al., 2025)). Higher values of $\mathrm{detect}(q(p(\cdot), \tau))$ means more verification-generation chains. We see that as this metric goes up, the performance (pass@$k$) increases. This verifies that the verification-generation gap (as measured by our Definition 4.1) is present even in largest open source state-of-the-art models.

One might ask what happens if we scale the model capacity even further; would we still see a verification-generation asymmetry being exploited by the model? We believe that the answer to this is conceptually yes as long as the model is pre-trained to imitate human-generated behavior from the Internet. In particular, to conceptually understand why asymmetries (Definition 4.1) arise at all in the first place, note that intuitively our definition measures how "easy" it is to get high reward with chaining skills vs. not. For any bounded capacity system, such a gap should exist as more "information" is gained by performing more computation via skill chaining on a given test input query as opposed to directly querying the system for an answer, with minimal or no test-time compute. With this intuition, such a gap exists for humans because humans don't have infinite computational and reasoning capacity and are themselves bounded capacity. Since we train models on human generated data during pre-training, this would naturally yield models that imitate systems of bounded capacity.

Of course, the set of test queries on which this kind of a gap is more prevalent will change depending upon the characteristics of the bounded capacity system: for instance, for previous gold-medal winners on the international math olympiad, verification and generation are likely equally complex on problems at the difficulty of AIME, whereas for a high-school student without a mathematical background there might still be a verification-generation asymmetry on an AIME question. This trend is also expected to be reflected in language models, and will be determined by the interaction between the base model size, pre-training data, and the test query. In principle though, unless all test queries are perfectly answered by a language model, such a gap should continue to exist on some test queries. We showed this concretely above using experiments with DeepSeek-R1 and Qwen3-8B models that both of them exhibit a verification-generation asymmetry despite being much bigger that our 1.7B model, but of course the set of test problems changed in each case.

## F  ASYMMETRIES BEYOND VG GAP: SUMMARIZATION VS. GENERATION

In Sec. 4, we present "chained asymmetries" as chaining of skills such as verification and generation. However, this is not the only notion of asymmetry. To illustrate a different form, we discuss results from a concurrent work Qu et al. (2025a). In this work, authors train a model to generate hints based on *summaries* of many rollouts on a problem, then conditioned on this hint generator they train a policy using RL to *generate* traces that optimize the typical outcome rewards.

Based on terminnology we introduce in Sec. 4, our interpretation of Qu et al. (2025a) is that they introduce an asymmetry between *planning or abstraction prediction* and *response generation*. In particular, they show that an LLM can improve performance on reasoning tasks by chaining two skills: (1) abstractly planning or generating high-level hints/hypotheses, followed by (2) generating the solution conditioned on one of the hints, as opposed to directly generating the answer. Imbuing this structure leads to improved performance, especially in regimes where pass@$k$ has saturated at large $k$ for a model trained with standard RL. Here, we include a subset of results from Qu et al. (2025a).

First, they evaluate GPT-4o-mini on several classification tasks, both with and without asymmetry (adding hints derived from summarized thinking traces). Performance improves significantly when hints are included.

| Model | Breast Cancer | Tweet Hate | Lobbying | Bank Note |
|---|---|---|---|---|
| GPT-4o-mini | 45% | 60% | 88% | 45% |
| GPT-4o-mini + asymmetry | **90%** | **90%** | **94%** | **100%** |

Table 6: Impact of chaining hint generation with response generation on classification tasks.

Next, they compare RL-trained models. The baseline is trained to generate final answers directly, while the asymmetry model is trained to chain hint generation and solution generation. For fairness, they allocate the same compute budget: the baseline uses $k$ rollouts, while the asymmetry model uses $\sqrt{k}$ rollouts for hints and $\sqrt{k}$ per hint for solution generation. On AIME 2025, the chained asymmetry again yields consistent improvements.

| Model | pass@4 | pass@16 | pass@64 |
|---|---|---|---|
| RL baseline (no asymmetry) | 51.1 | 65.2 | 77.1 |
| RL (with asymmetry) | **59.0** | **71.1** | **80.5** |

Table 7: AIME 2025 results comparing RL baseline with RL using chained asymmetry.

The above results indicate, that there other skills like summarization, retrieval, instruction following in the base pre-trained model, which can be chained with generation to improve final performance. With the right base model, and curricula, RL can amplify all of the "chained asymmetries" that improves training rewards, incentivizing in-context exploration and as a consequence also improve the extrapolation performance.

# G  ADDITIONAL EXPERIMENTS AND DETAILS FOR SECTION 5 (NEGATIVE GRADIENT)

## G.1  DETAILS FOR CDOWN

We trained models for 90 steps on problems with 5 candidate numbers with a training budget of 2k.

**Cumulative unique attempts plot.** Fig. 6 (left) was filtered on incorrect traces on problems with ¡ 50% success across gradient steps. We select only incorrect traces to capture the ability of the model to explore for the correct trace, rather than to output diverse correct traces once one is found. We filter for problems with ¡ 50% success across training for GRPO and GRPOMask because otherwise the algorithm with better rewards would see more problems with lower cumulative unique attempts, as the correct traces are discovered early and subsequently reinforced.

**Evolution of the conditional distribution given past attempts in CDOWN.** We run ablations on the conditional distribution of a new attempt (sequence of tokens that constitute an attempt to plug-in operations so as to match the target CDOWN) given past attempts in three different settings, shown in Fig. 15. In (a), we plot $\log p(a_k|a_{1:k-1}) - \log p(a_k|a_{1:k-2})$, which should average to roughly 0 if the attempts are independent. As training progresses, this quantity grows, indicating a correlation between attempts, especially with larger $k$ (potentially because the new attempt can attend to more previous attempts, and thus becomes more dependent on them). In (b), we plot $\log p(a_k|a_{1:k-1}) - \log p(a_{k-1}|a_{1:k-2})$, which also grows over time. This indicates that the conditional distribution $p(\text{new attempt}|\text{past attempts})$ sharpens as the number of past attempts grows, implying that the model gets more confident as it explores more in-context. In (c), we plot $\log p(a_{k-1}|a_{1:k-1})$ and note that it reduces with more attempts way more on the trained model, compared to initialization. This means, that the model has learned not to repeat its previous attempt when it immidiately re-attempts to solve the problem. These three trends jointly tell us that the learned model indeed learns to explore-in-context where it adapts and sharpens the conditional distribution over the next attempt with more previous attempts.

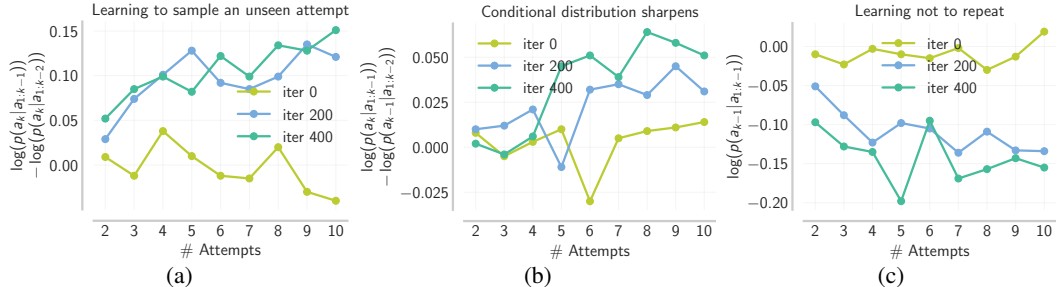

Figure 15: *Probing the conditional distributions conditioned on past attempts in* CDOWN. **(a):** New attempts are not independent of past attempts **(b):** Model becomes more certain of what to try next given more past attempts **(c):** Model learns not to repeat past attempts.

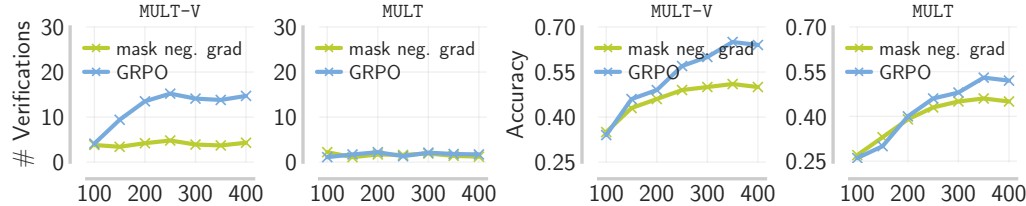

Figure 16: *Negative gradient amplifies verification when VG gap is large.* While utilizing the negative gradient amplifies the number of calls to verification in MULT-V, the number of verification calls does not grow over training in MULT. Interestingly, though, we find that when negative gradient is masked out on MULT-V, the number of verification calls is still very low and does not increase, corroborating our findings that exploration driven by negative gradients results in in-context exploration only in the presence of asymmetries in the base model. A similar trend is also observed in terms of the raw accuracy.

## G.2 ADDITIONAL EXPERIMENTS WITH MULT

In Section 5 we saw that training with the negative gradient leads to more exploration during RL training, which in turn leads to the amplification of any chained asymmetries that may be present in the base model, *e.g.*, more generation-verification steps. In particular, we noted the increase in the number of verification steps in Fig. 5(b). To see how negative gradients affect the response length and number of chained asymmetries in the absence of a strong VG gap, we compare running GRPO with and without negative gradients on our multiplication task MULT where the VG gap is weaker in the base model.

We plot results in Fig. 16, where we note two trends when running RL training with and without negative gradients on MULT (without VG gap), and MULT-V (with VG gap) using a training budget of 4096 tokens. First, we note that the number of verifications is higher when we use negative gradients in a setting with a large VG gap. When the VG gap is absent, the number of chained asymmetries (verification-generation steps) are roughly the same with and without masking the negative gradient. Second, we note that the accuracy is much higher with negative gradients in the presence of VG gap (MULT-V), and comparable to a run where we mask the negative gradients in the setting where the VG gap is poor (MULT). Together, this tells us that the boost in exploration driven by negative gradients leads to more chained asymmetries when the base model presents some of them, like a large VG gap.

## G.3 GRADIENT NORM OF GRPO VS. GRPO WITH MASKED NEGATIVE GRADIENTS.

In Fig. 17, we compare the $\ell_2$ gradient norm during training for standard GRPO and our masked-GRPO variant on DMATH. We observe that masking negative gradients produces a larger gradient norm over training steps, whereas standard GRPO exhibits a smaller one. This is mainly because we only average the positive gradient over the tokens with a positive advantage (since they end in a correct response) for masked GRPO, but in the std. GRPO setting the gradient is averaged over all tokens (including the zero-advantage tokens), which can result in a lower gradient norm for std.

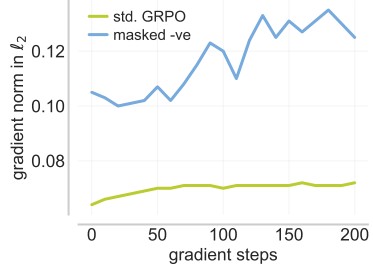

Figure 17: **Gradient norm of Masked GRPO vs. GRPO**: We plot the gradient norm for our experiment in Sec. 5 where we run standard GRPO and the masked version of GRPO on DMATH and find that masked GRPO observes a higher gradient norm. This is mainly because we only average the positive gradient over the tokens with a positive advantage (since they end in a correct response) for masked GRPO, but in the std. GRPO setting the gradient is averaged over all tokens (including the zero-advantage tokens), which can result in a lower gradient norm for std. GRPO.

GRPO. This gap in performance between masked-GRPO and GRPO is mainly from the lack of exploration (lower entropy and verification-generation chains) as we note in Fig. 5 and our theoretical analysis in Appendix A, and has less to do with the magnitude of the gradient update.

## G.4  ANALYZING THE PERFORMANCE DROP FROM MASKED NEGATIVE GRADIENT

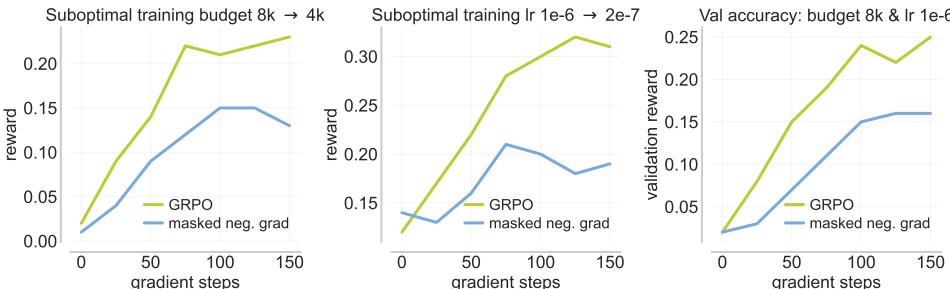

Figure 18: *Performance drop from masking the negative gradient is orthogonal to the drop from choosing suboptimal hyperparameters:* We plot the performance of GRPO and GRPO where we mask the negative gradient in two additional settings (beyond what we plot in Fig. 5). In particular, we change the hyper-parameters for the GRPO run on DMATH to suboptimal configurations: dropping the token budget from 8k to 4k and learning rate from 1e-6 to 2e-7. With all else being the same, we compare the training rewards of GRPO with masked GRPO. In addition, we also plot the validation performance of masked GRPO (run with the configurations in Fig. 5: budget of 8k and learning rate of 1e-6) to show that masked GRPO indeed generalizes. This suggests that when we are masking the gradient, we are not necessarily training the model in ways where it is expected to fail.

In Fig. 18 we compare standard GRPO (with both positive and negative gradients) to masked GRPO (only positive gradients) on DMATH under two deliberately suboptimal configurations: (i) reducing the training token budget from 8k to 4k, and (ii) reducing the learning rate from $1 \times 10^{-6}$ to $2 \times 10^{-7}$. These settings are expected to degrade the performance of standard GRPO, allowing us to ask whether the additional drop observed with masking can be explained purely by hyperparameter choices. Holding everything else fixed, we find that masked GRPO consistently underperforms standard GRPO and exhibits the same qualitative training dynamics as in Fig. 5: lower reward, lower entropy, and shorter output lengths. We also report validation accuracy for masked GRPO under the original (8k, $1 \times 10^{-6}$) configuration, confirming that the masked policy still generalizes. Taken together, these results show that (i) masking negative gradients reliably reduces exploration (entropy and trajectory length) and harms performance across a range of hyperparameters, and (ii) the performance drop from masking is *orthogonal* to the degradation induced by suboptimal hyperparameter choices.

# H   ILLUSTRATING THE EFFICACY OF EQ. 2 AND COUPLED CURRICULUM

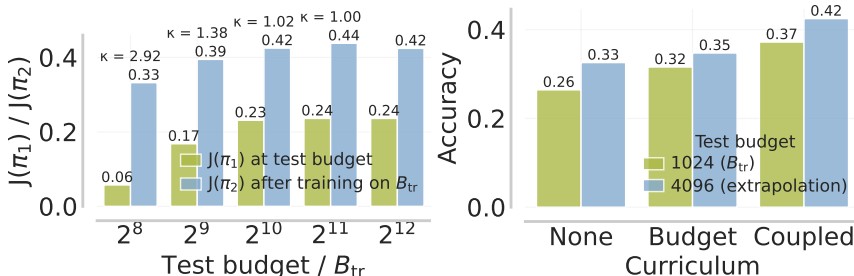

Figure 19: **Left:** Given a model $\pi_1$, at end of stage 1 of the e3 curriculum on CDOWN, we illustrate how choosing budget $B_{tr}$ for stage 2 of the curriculum via our presribed Eq. 2 is optimal. On the x-axis we plot the test budget for evaluating $\pi_1$. Based on the evaluation of $\pi_1$, we find that $2^{10}$ tokens is the smallest token budget with $\kappa < 1.2$. Based on Eq. 2, we now set $2^{10}$ as the token budget for stage 2 training of the curriculum, and find that $\pi_2$ trained at 1024 tokens is comparable with any $\pi_2$ trained at a higher budget value, and much better than $\pi_2$ trained at a lower value. $J(\pi_1)$ is pass@128 performance and $J(\pi_2)$ is pass@1 accuracy. **Right:** *e3 on* CDOWN*:* our coupled curriculum performs better than budget currciulum ($2^8 \rightarrow 2^{10}$) or no curriculum at all.

In this section, we show that for a given dataset, picking the training token budget based on Eq. 2 actually results in the best performance of the RL trained model. We demonstrate this on CDOWN by training on problems of 3, 4, and 7 numbers. In our coupled curriculum, we first train on problems of easier difficulty with 3 and 4 numbers, on a relatively smaller token budget of 256 tokens. Next, we use Eq. 2 to pick the training budget for stage 2 of the curriculum where we train only on difficult problems (CDOWN problems with 7 numbers). Following Eq. 2 with say $\kappa = 1.2$, to select the budget for the next stage, we examine the performance of this first-stage model on the second-stage dataset consisting of harder problems with 7 numbers. Eq. 2 prescribes that we pick the smallest but reasonable $B$ such that there is only a marginal improvement from extending $B$ to $2B$, defined by $\kappa = 1.2$. As shown in Fig 19 (left), this corresponds to $B_{tr,1}^{\star}(D_1) = 1024$ (where $D_1$ is the second stage training dataset). Indeed, Fig 19 (left) shows that when we train the second stage model at $B_{tr,1}^{\star}(D_1) = 1024$, we get nearly the best performance of 0.42 on the harder problems (7 numbers). We also note that while $B_{tr} = 2048$ marginally improves test performance over $B_{tr} = 1024$, it is unclear apriori if $B_{tr} = 2048$ would train stably. Our goal here is to come up with a prescriptive thumb rule that can inform the practitioner of the smallest training budget at which they can expect to train a model on a given dataset and hope to see the performance of the trained model extrapolate well with additional test-time tokens. We also find in the figure on the right, that our coupled curriculum outperforms budget curriculum or not training with any curriculum.

**Choice of $\kappa$.** In Fig. 19(left), if we had picked a value of $\kappa > 1.3$ we would have run RL training on a much smaller and suboptimal token budget for stage 2 of the curriculum. At the same time, if we had picked $\kappa < 1.05$, we would have picked an unnecessarily large training budget.

# I   ADDITIONAL EXPERIMENTS AND DETAILS FOR SECTION 6 (CURRICULA TRAINING)

For easy-to-see comparisons of e3-1.7B with open other open source models of similar and bigger sizes, we split the plot in Fig. 8(a,b) into four different plots, two each on AIME 2025 and HMMT 2025. In each set, we compare e3-1.7B separately with models of size <2B and >2B (Fig. 20)

## I.1   TRAINING DETAILS AND IN-DISTRIBUTION PERFORMANCE ON TRAINING BUDGET

We present our hyperparameters for e3 training runs in Table 8.

**Note on in-distribution performance.** In Sec. 6 we note that for best extrapolation performance, it is important to vary the mixture of tasks in the dataset, as well as the training budget (max token length) in a a coupled way, over the course of RL training. Here, we note that if we were to only care about in-distribution performance, i.e., performance on a fixed task mixture (of equally proportioned

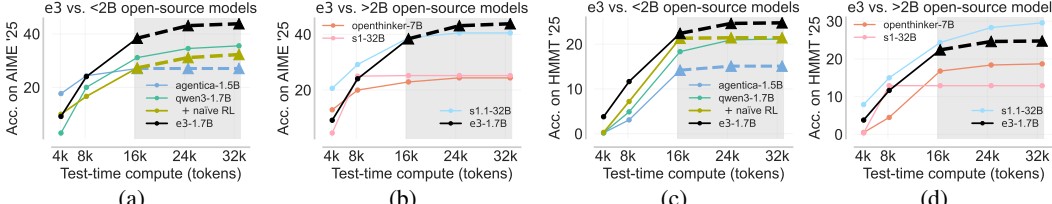

Figure 20: *Comparing **e3** with open source models.* In the above figure, the shaded area indicates the extrapolation regime. e3-1.7B trained with our RL recipe achieves *state-of-the-art* performance across models <2B and is comparable or better than some 7B and 32B models trained with supervised fine-tuning.

easy, medium, and hard questions in DMATH), then the best way to train is to match the test token budget and prompt mixture with training. In particular, training only on easy problems and a budget of 8k yields a performance of $54.3\%$ on a test dataset consisting of all tasks (from easy, medium and hard splits). But, if we match the test mixture with train, and train on all difficulties, then on the same 8k test budget, we note a performance of $58.9\%$, averaged over all difficulties. Note that the exptrapolation performance (on hard, out-of-distribution AIME '25 questions) of the same models is flipped in Fig. 7, indicating that the curricula design is mainly needed for extrapolation, via in-context exploration, as opposed to best performance at a fixed test $B_{\text{tr}}$.

| Hyperparameter | Values ($B_{\text{tr}} = 8$k) | Values ($B_{\text{tr}} = 16$k) |
|---|---|---|
| train_batch_size | 128 | 64 |
| ppo_mini_batch_size | 32 | 32 |
| learning_rate | 1.0e-6 | 1.0e-6 |
| kl_loss_coef | 0.001 | 0.001 |
| entropy_coeff | 0.002 | 0.001 |
| temperature | 0.6 | 0.6 |
| rollout.n | 8 | 32 |
| ppo_lowerclip_threshold | 0.2 | 0.2 |
| ppo_higherclip_threshold | 0.5 | 0.35 |

Table 8: Verl Sheng et al. (2024) hyperparameters used for e3 runs on DMATH.

**Hyperparameters for e3.** In Tab 8, we report the hyperparameters we used for RL training during different stages of our coupled curriculum runs on DMATH. For the first stage of training on $B_{\text{tr}}$=8k, we used fewer number of rollouts per prompt (16), and increased this to 32 for second stage training when $B_{\text{tr}}$=16k. We did this to account for the larger outcome-reward variance typically associated with long horizon RL training (Agarwal et al., 2021). In general, we find that a successful RL run at a budget of 16k can generally characterized by the following trends: **(i)** averag per-token entropy increases during training (or at least does not drop during training); and **(ii)** the number of chained asymmetries (verification attempts) and the response length increases during RL training. To improve token-level entropy we move away from purely on policy RL, and use off-policy data to update the current policy, with the policy density ratio clipping mechanism to avoid aggressively off-policy updates. Consistent with the findings in Yu et al. (2025), we find that when updating the policy on stale off-policy data, using a higher clip ratio for the positive advantage tokens is critical for increasing token-level entropy during RL training. This is mainly to weight the probability of some very low probability and positive advantage tokens. But increasing the clip ratio too substantially can also de-stabilize training, as we observed in the 16k training runs, due to which we dropped the clip threshold from 0.5 to 0.35. The rest of the hyperparameters are consistent with the default options in Verl (Sheng et al., 2024).

**Evaluation protocol for the "Wait" extrapolation method in Muennighoff et al. (2025).** In Fig. 9, we explicitly intervene on the model generations (except for our model e3-1.7B) with the "Wait" extrapolation technique from Muennighoff et al. 2025 and plot the performance against the average length of the extrapolated trace (on the x-axis). Concretely, we first run a 16K-token rollout, stopping at </think> instead of EOS. To extend it with wait, we repeatedly append a rotating prefix from ["Wait", "But wait", "Alternatively", "But hold on"], rollout again (stopping at </think> or 2048 tokens), and after 2 to 8 such extensions we let the model finish without stopping at </think>.

**Classifying problem difficulty on DMATH.** On the DMATH task, we classify a problem as easy/hard based on the pass@1 performance of QwQ-32B (more capable than our base model Qwen3-1.7B) on a given problem. For example, as we describe in Section 5, we bucket a problem in DMATH (sourced from DeepScaleR (Luo et al., 2025b)) as hard if the pass@1 performance (measured on 32 rollouts) of QwQ-32B is 0. We classify it as medium if this value is between 0.0 and 0.2 and easy otherwise.

## I.2 FIXED TRAIN BUDGET, VARY DATASET CURRICULUM ON CDOWN

In this subsection, we demonstrate that training with a data curriculum based on difficulty with a fixed train budget can lead to over-exploratory output traces, on the example task of CDOWN. With the data curriculum (i.e., fixed budget, vary data), we train first on CDOWN problems with 3 candidate numbers (the "easy" problems) for 60 gradient steps, then those with 6 candidate numbers for 60 gradient steps (the "hard" problems), with a 1k budget across all steps. We compare this with the coupled curriculum in which the first 60 gradient steps are trained with a budget of 256. As shown in Fig. 21, the latter achieves better reward on "hard problems".

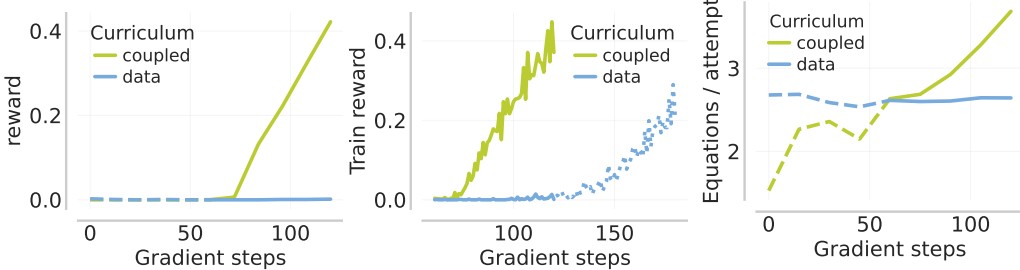

Figure 21: **Coupled vs. data curriculum on CDOWN:** training only on easy problems at large budgets leads to overfitting on "over exploratory" traces, failing to balance explore-exploit tradeoff on harder problems later on. Reward graphs are displayed for hard problems.

***Why is data curriculum worse than the coupled curriculum?*** We can view the learning of correct traces as largely composed of two stages: (i) negative gradients encourage exploration, leading to the discovery of correct traces, (ii) positive gradients reinforce correct traces, once discovered.

For (i), we observe that training on easy problems exacerbates a tendency to perform over-exploratory in-context exploration "underthinks" (see Example 3 in App. N), restricting the discovery of solutions to harder problems. When utilizing a coupled curriculum, this bias propagates to a shorter budget when compared to the data curriculum, since easy problems are trained on 256 rather than 1K tokens. As shown in Figure 21, the average number of equations per attempt (naïvely, with 3 candidate numbers, 2 equations are required to perform a complete attempt vs. 5 equations for 6 candidates) increases noticeably for the coupled curriculum in the second stage, but plateaus for the data curriculum, implying overfitting on "over-exploratory" traces during the first stage.

Furthermore, for (ii), even when nontrivial positive rewards are obtained as we run the data curriculum on hard problems for 60 additional steps (steps 120 to 180), the training reward curve converges more slowly compared to the coupled curriculum (steps 60 to 120), implying that the data curriculum is also worse at reinforcing correct traces if the behavior is over exploratory. While we do not run many controlled experiments to identify why this might be the case, we hypothesize that this is because of imperfect and noisy credit assignment on over-exploratory traces with outcome rewards. It is unclear which segments of the trace should be reinforced vs which segments might simply confuse the model.

## I.3 COMPARING EXTRAPOLATION OF NAÏVE RL WITH E3

In Fig. 22, we compare e3-1.7B, trained with our coupled data-and-budget curriculum on DMATH (easy problems at 8k, followed by medium and hard problems at 16k), to a naïve RL baseline trained on the same dataset but only at a fixed 16k-token budget. While the naïve baseline shows only mild gains and quickly saturates once we extrapolate beyond its training budget, e3 continues to improve

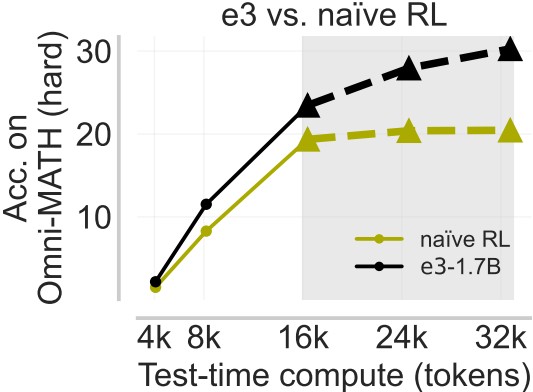

**Figure 22:** **Extrapolation performance of `e3` vs. naïve RL on hard problems:** We plot the performance of `e3`-1.7B trained with our coupled curriculum (Section 6) on DMATH, with a model that is trained directly at 16k training token budget on the same dataset (shaded region indicates the extrapolation regime). We find that `e3` extrapolates test-time compute better than the naïve RL baseline, when evaluated on a test set of hard problems from Omni-MATH (Gao et al., 2024) dataset.

as we increase test-time compute to 24k and 32k tokens, achieving substantially higher accuracy on the hard split of Omni-MATH problems: `e3`-1.7B outperforms the baseline by 0.07 points (gap of 7%) at the training budget 16k tokens, and this gap grows to 0.11 points (11%) when we evaluate the models on the extrapolated budget of 32k tokens. In Fig. 8, we compare the extrapolation performance of these two methods on AIME and HMMT 2025. There too we see `e3` perform better than the naïve RL baseline but the gains are already large on the training budget of 16k tokens. In the above figure, we see the gap widen as we allow a bigger test time token budget on a harder set of 64 problems (sampled IID from Omni-Math levels 6-8).

## J  OMITTED PROOFS

In this section, we present the formal version of Theorem A.1, and provide a detailed proof for it. First, we introduce some notations and provide a proof overview.

**Notations.** We use the shorthand $H(M; \mathbf{s})$ to denote the entropy of the conditional distribution over the next action $a_{t+1}$ given the current state $\mathbf{s}$. We also use $M^{(i)}$ to refer to the policy parameters (for the softmax policy in Eq. 3) at iteration $i$ of RL training, and use the shorthand $\pi^{(i)}$ to denote the policy induced by the parameter $M^{(i)}$. We use $\nabla_{M^{(i)}} f(M^{(i)})$ to denote the gradient of function $f(M)$, with respect to $M$, evaluated at $M = M^{(i)}$. Finally, we use $M_\mathbf{s}$ to denote the row of softmax parameters that model the distribution $\pi_M(\cdot \mid \mathbf{s})$, *i.e.*, the row of parameters $M(\cdot \mid \mathbf{s})$ in our parameter matrix $M$.

**Proof overview.** Without loss of generality, we fix an arbitrary state $\mathbf{s}$ that is different from `stop`. Given the parameters $M^{(i)}$ at current RL iterate $i$, we do a Taylor expansion of $H(M^{(i)}; \mathbf{s})$ around $M^{(i)}$, and then show that the gradient $\nabla_{M^{(i)}} H(M^{(i)}; \mathbf{s})$ is positively correlated with the policy gradient with high probability over the sampling of the action $a \sim \pi_{M^{(i)}}(\cdot \mid \mathbf{s})$, *i.e.*:

$$\langle \nabla_{M_i} H(M^{(i)}; \mathbf{s}) , \ \nabla_{M^{(i)}} \log \pi(a \mid \mathbf{s}) \ A(\mathbf{s}, a) \rangle \geq 0, \tag{4}$$
$$\text{whp. over sampling of action } a \sim \pi_{M^{(i)}}(a \mid \mathbf{s})$$

Before, we prove our result that lower bounds the increase in entropy with negative gradients, we present derivations of the entropy gradient with respect to the model parameters, as well as the policy gradient, which will simplify some calculations in the proof.

**Lemma J.1** (Entropy gradient for the softmax bi–gram conditional). *Fix a previous action (because the bi–gram state is $s_t = a_{t-1}$, conditioning on the state is equivalent to conditioning on the last action) $a \in \mathcal{A}$. Let the (column-wise) logit matrix at time $t$ be $M \in \mathbb{R}^{(K+1) \times K}$, and define the*

*corresponding softmax conditional distribution*

$$\pi_M(a^+ \mid a) = \frac{\exp\big(M(a^+ \mid a)\big)}{Z(a)}, \qquad Z(a) = \sum_{a' \in \bar{A}} \exp\big(M(a' \mid a)\big). \tag{5}$$

*Let the Shannon entropy of this conditional distribution be $H\big(\pi_M(\cdot \mid a)\big)$ or $H(M \mid a)$ Then $\nabla_M H(M \mid a) \in \mathbb{R}^{K+1}$ is given by:*

$$\nabla_M H(M \mid a) = = -\pi \odot \big(\log \pi + H(\pi)\,\mathbf{1}\big) = -\big[\pi_i\big(\log \pi_i + H(\pi)\big)\big]_{i \in \bar{A}}, \tag{6}$$

*Proof.* Write $p_{a^+} := \pi_M(a^+ \mid a)$ for brevity. By definition of the entropy,

$$H = -\sum_{a^+} p_{a^+} \log p_{a^+}. \tag{7}$$

Insert the softmax expression:

$$\log p_{a^+} = M(a^+ \mid a) - \log Z(a). \tag{8}$$

Hence,

$$H = -\sum_{a^+} p_{a^+} \big[M(a^+ \mid a) - \log Z(a)\big] \tag{9}$$

$$= -\sum_{a^+} p_{a^+} M(a^+ \mid a) + \log Z(a) \underbrace{\sum_{a^+} p_{a^+}}_{=1}. \tag{10}$$

Rearranging yields the following closed form expression:

$$H = \log Z(a) - \sum_{a^+} p_{a^+} M(a^+ \mid a). \tag{11}$$

Computing the Jacobian of the softmax we get:

$$\frac{\partial \pi_i}{\partial M(j \mid a)} = \pi_i\big(\delta_{ij} - \pi_j\big), \qquad J := \nabla_{M(\cdot \mid a)} \pi = \mathrm{diag}(\pi) - \pi \pi^\top. \tag{12}$$

Starting from the definition $H = -\sum_i \pi_i \log \pi_i$ and using the chain rule,

$$\frac{\partial H}{\partial M(j \mid a)} = -\sum_i \frac{\partial \pi_i}{\partial M(j \mid a)}(1 + \log \pi_i) = -\sum_i \pi_i(\delta_{ij} - \pi_j)(1 + \log \pi_i). \tag{13}$$

Separating the term $i = j$ from the rest:

$$\frac{\partial H}{\partial M(j \mid a)} = -\pi_j(1 - \pi_j)(1 + \log \pi_j) + \pi_j \sum_{i \neq j} \pi_i(1 + \log \pi_i) \tag{14}$$

$$= \pi_j \Big[\sum_i \pi_i(1 + \log \pi_i) - (1 + \log \pi_j)\Big]. \tag{15}$$

Because $\sum_i \pi_i(1 + \log \pi_i) = 1 + \sum_i \pi_i \log \pi_i = 1 - H(\pi)$, we obtain

$$\frac{\partial H}{\partial M(j \mid a)} = \pi_j\big(1 - H(\pi) - 1 - \log \pi_j\big) = -\pi_j\big(\log \pi_j + H(\pi)\big), \tag{16}$$

which gives the stated component-wise form. Writing this for every $j$ simultaneously yields the vector expression with the Hadamard product.

$\square$

**Lemma J.2** (Policy gradient for the conditional distribution). *For an action $a \sim \pi_M(\cdot \mid \mathbf{s})$, sampled from a policy $\pi_M(\cdot \mid \mathbf{s})$, at state $\mathbf{s}$, the policy gradient is given by: $\nabla_{M_\mathbf{s}} \log \pi(a \mid \mathbf{s}) \cdot A(\mathbf{s}, a)$, where $A(\mathbf{s}, a)$ is the advantage of action $a$. The expression for the $b^{th}$ coordinate of the policy gradient can be written down in closed form as:*

$$[\nabla_{M_\mathbf{s}} \log \pi(a \mid \mathbf{s}) \cdot A(\mathbf{s}, a)]_b = (\mathbf{1}(b = a) - \pi(a \mid \mathbf{s})) \cdot A(\mathbf{s}, a),$$

*where $\mathbf{1}(\cdot)$ is an indicator function.*

*Proof.* Write $Z := \sum_c \exp M(c \mid \mathbf{s})$ and $\pi_b := \pi_M(b \mid \mathbf{s}) = \exp M(b \mid \mathbf{s})/Z$ for brevity. By definition

$$\log \pi_M(a \mid \mathbf{s}) = M(a \mid \mathbf{s}) - \log Z. \tag{17}$$

For any coordinate $b \in \bar{\mathcal{A}}$,

$$\frac{\partial}{\partial M(b \mid \mathbf{s})} \log \pi_M(a \mid \mathbf{s}) = \underbrace{\mathbf{1}(b = a)}_{\text{derivative of } M(a|\mathbf{s})} - \frac{1}{Z} \frac{\partial Z}{\partial M(b \mid \mathbf{s})}$$

$$= \mathbf{1}(b = a) - \frac{\exp M(b \mid \mathbf{s})}{Z} = \mathbf{1}(b = a) - \pi_b. \tag{18}$$

Multiplying every coordinate by the common scalar $A(\mathbf{s}, a)$ produces the stated expression for $g(\mathbf{s}, a; M)$. $\qquad\square$

**Theorem J.3** (Negative gradient increases $H(M; \mathbf{s})$ when $p(a^\star|\mathbf{s})$ is low). *For any state $\mathbf{s}$, current parameters $M^{(i)}$, suppose the most likely action $\bar{a}$ is incorrect, i.e., $a^\star \neq \bar{a} =: \arg\max_b \pi_{M^{(i)}}(b \mid \mathbf{s})$, where the probability of sampling $\bar{a} \mid \mathbf{s}$ is $\pi_{\bar{a}}$, and the second most likely action has probability $\pi_{\bar{a}} - \varepsilon$. Then, for a small enough learning rate $\eta > 0$ s.t. with probability $\geq \pi_{\bar{a}}$, negative gradient produces $\pi^{(i+1)}$ with entropy $H(M^{(i+1)}; \mathbf{s}) > H(M^{(i)}; \mathbf{s})$. Additionally, there exists a universal constant $c > 0$ s.t., $H(M^{(i+1)}; \mathbf{s}) - H(M^{(i)}; \mathbf{s}) \geq c\eta \cdot K\varepsilon^2(1 - p_{\bar{a}})$ whenever $\pi_{\bar{a}} \geq \varepsilon + e^{-H(M^{(i)}; \mathbf{s})}$. In contrast, without negative gradient the entropy remains same with probability $1 - \pi(a^\star \mid \mathbf{s})$.*

*Proof.* For simplicity let us denote $\pi^{(i)} = (\pi_1, \ldots, \pi_{K+1}) \in \Delta(\bar{\mathcal{A}})$ be the conditional distribution produced by a bi-gram softmax column $\pi_{M^{(i)}}(\cdot \mid \mathbf{s})$, i.e., the probability of sampling action $a$ at state $\mathbf{s}$, with model parameters given by the current RL iterate $M^{(i)}$. Let us also denote,

$$\bar{a} = \arg\max_i \pi_i, \qquad H(M^{(i)}; \mathbf{s}) =: -\sum_{a \in \bar{\mathcal{A}}} \pi_a \cdot \log \pi_a,$$

where $\pi_a$ is the probability of sampling action $a$ at state $\mathbf{s}$. Given that the current policy $\pi_M$ samples action $a \sim \pi^{(i)}(\cdot \mid \mathbf{s})$, the stochastic policy gradient that updates the parameter is given by:

$$M_{\mathbf{s}}^{(i+1)} = M_{\mathbf{s}}^{(i)} + \eta \, \nabla_{M_{\mathbf{s}}^{(i)}} \log(\pi^{(i)}(a \mid \mathbf{s})) \cdot A(\mathbf{s}, a), \tag{19}$$

where $\eta$ is the learning rate. Note, that the policy parameters would only be updated for the row corresponding to the state $\mathbf{s}$. For simplicity, let us use the notation $g$ for:

$$g =: \nabla_{M_{\mathbf{s}}^{(i)}} \log(\pi^{(i)}(a \mid \mathbf{s})) \cdot A(\mathbf{s}, a). \tag{20}$$

Then, $M_{\mathbf{s}}^{(i+1)} - M_{\mathbf{s}}^{(i)} = \eta \cdot g$. A second–order Taylor expansion of the concave function $H(M; \mathbf{s})$ gives, for some $\tilde{M}$ on the segment $[M^{(i)}, M^{(i+1)}]$:

$$H(M^{(i+1)}; \mathbf{s}) = H(M^{(i)}; \mathbf{s}) + \eta \cdot \langle \nabla_{M^{(i)}} H(M^{(i)}; \mathbf{s}), g \rangle$$

$$+ \frac{\eta^2}{2} \cdot (g)^\top \nabla_{\tilde{M}_{\mathbf{s}}}^2 H(\tilde{M}; \mathbf{s}) \, (g). \tag{21}$$

Let the least eigenvalue of the Hessian of the conditional entropy (note that the entropy is a concave function) with respect to the logits be $\rho_{\tilde{M}_{\mathbf{s}}}$, and $|\rho_{\tilde{M}_{\mathbf{s}}}| < \infty$, the moment $\pi^{(i)}(a \mid \mathbf{s}) > 0$ for all actions $a \in \bar{\mathcal{A}}$. This condition is easily satisfied by any policy in our policy class, with finite values of the parameter matrix $M$. Thus, whenever $\langle g, \nabla_{M_{\mathbf{s}}^{(i)}} H(M^{(i)}; \mathbf{s}) \rangle > 0$ there exists a small enough learning rate $\eta$,

$$\eta \leq \frac{2\langle g, \nabla_{M_{\mathbf{s}}^{(i)}} H(M^{(i)}; \mathbf{s}) \rangle}{\rho \|g\|_2^2}, \tag{22}$$

such that $H(M^{(i+1)}; \mathbf{s}) - H(M^{(i)}; \mathbf{s})$ is strictly positive. Thus, we can continue to reduce learning rate $\eta$ such that we can ignore $\mathcal{O}(\eta^2)$ terms in Eq. 21, to get the bound:

$$H(M^{(i+1)}; \mathbf{s}) - H(M^{(i)}; \mathbf{s}) \geq \frac{\eta}{2} \cdot \langle \nabla_{M_{\mathbf{s}}^{(i)}} H(M^{(i)}; \mathbf{s}), \nabla_{M_{\mathbf{s}}^{(i)}} \log(\pi^{(i)}(a \mid \mathbf{s}) \cdot A(\mathbf{s}, a)) \rangle \tag{23}$$

Next, it remains to bound the right hand side of Eq. 23 with high probability over the sampling of the action $a$. For a single incorrect action draw $a \sim \pi$ we set $A(\mathbf{s}, a)$ to be $-1$ and for such an incorrect action we define the alignment scalar:

$$\mathcal{T}(a) \;=: \; -\Big\langle \nabla_{M_{\mathbf{s}}^{(i)}} \log \pi^{(i)}(a \mid \mathbf{s}) \cdot A(\mathbf{s}, a), \; \nabla_{M_{\mathbf{s}}^{(i)}} H(M^{(i)}; \mathbf{s}) \Big\rangle \tag{24}$$

Plugging in the derivation of $\nabla_{M^{(i)}} H(M^{(i)}; \mathbf{s})$ from Lemma J.1, we compute the closed form expression for $T(a_i)$ using the following definitions:

$$v_i =: \pi_i \big( H(M^{(i)}; \mathbf{s}) + \log \pi_i \big) \quad \text{and,} \quad \mu =: \sum_{a \in \bar{\mathcal{A}}} \pi_a v_a \tag{25}$$

Thus, one has $T(a)$ satisfy:

$$\mathcal{T}(a) = v_a - \mu \qquad \text{when,} \; a \in \bar{\mathcal{A}}, \; i \neq a^{\star}. \tag{26}$$

Note that $v_i$ is an increasing function in $\pi_i$ whenever $\pi_i > e^{-H(M^{(i)}; \mathbf{s})}$. Next, we note that $v_{\bar{a}} \geq 0$.

$$\pi_{\bar{a}} \geq \frac{1}{|\bar{\mathcal{A}}|} \;\implies\; \pi_{\bar{a}} \geq e^{-H(M^{(i)}; \mathbf{s})} \quad \text{since,} \; H(M^{(i)}; \mathbf{s}) \leq \log |\bar{\mathcal{A}}| \;\implies\; v_{\bar{a}} \geq 0. \tag{27}$$

Finally, since $v(x) = x H(M^{(i)}; \mathbf{s}) + x \log x$ is convex in $x$:

$$v_{\bar{a}} \geq \sum_j \pi_j v_j \;\implies\; v_{\bar{a}} - \mu \geq 0 \tag{28}$$

The above two implications in Eq. 27 and Eq. 28, and the fact that $\bar{a} \neq a^{\star}$, together lead us to a deterministic lower bound on $T(\bar{a})$, implying that it is always positive:

$$\mathcal{T}(\bar{a}) \;\geq\; 0. \tag{29}$$

This completes the derivation for the first part of Theorem J.3, which does not assume anything about the conditional distribution $\pi^{((i))}(\cdot \mid \mathbf{s})$, directly yielding the following result.

**Result (i):** Under the conditional distribution $\pi^{(i)}(\cdot \mid \mathbf{s})$, whenever the most likely action $\bar{a} \neq a^{\star}$, then with probability at least $\pi_{\bar{a}}$, $T(a) \geq 0$, for $a \sim \pi^{(i)}(\cdot \mid \mathbf{s})$, and any policy $\pi$ in our class of softmax policies. Finally, we plug this into Eq. 23 to conclude that the policy gradient update with probability $\pi_{\bar{a}}$ always increases entropy, for a small enough learning rate.

Next, we lower bound $T(\bar{a})$ when the second most likely action under the distribution satisfies an additional condition. For this, let us fix some $\varepsilon \geq 0$, such that for $q = \arg\max_{b \neq \bar{a}} \pi^{(i)}(b \mid \mathbf{s})$, we have $\pi_q = \pi_{\bar{a}} - \varepsilon$. Based on our alignment scalar $\mathcal{T}(\cdot)$, we define the function $g(x)$ as follows:

$$g(x) \;=\; x \big( H(M^{(i)}; \mathbf{s}) + \log x \big), \qquad 0 < x \leq 1, \tag{30}$$

where $H(M^{(i)}; \mathbf{s})$ is the conditional entropy we defined previously. Then, given the most probable action $\bar{a}$, and the runner up action $q$, the gap between $\mathcal{T}(\bar{a})$ can be lower bounded down as follows when $\pi_q \geq \exp(-H(M^{(i)}; \mathbf{s}) - 1)$:

$$\mathcal{T}(\bar{a}) = g(\pi_{\bar{a}}) - \pi_{\bar{a}} \cdot g(\pi_{\bar{a}}) - \sum_{b \neq \bar{a}} \pi_g \cdot g(b)$$
$$\geq (1 - \pi_{\bar{a}}) \cdot g(\pi_{\bar{a}}) - (1 - \pi_{\bar{a}}) \cdot g(q) = (1 - \pi_{\bar{a}}) \cdot (g(\pi_{\bar{a}}) - g(\pi_q)), \tag{31}$$

where the second equality follows from the fact that $g(\pi_q) \geq g(b)$ for any $b \neq \bar{a}$ as soon as $\pi_q \geq \exp(-H(M^{(i)}; \mathbf{s}))$, which is implied by the condition on $\pi_{\bar{a}}, \varepsilon$ in Theorem J.3.

By the mean–value form of Taylor's theorem there exists a $\xi \in [\pi_q, \pi_{\bar{a}}]$ such that

$$g(\pi_{\bar{a}}) \;=\; g(q) \;+\; \varepsilon\, g'(q) \;+\; \frac{\varepsilon^2}{2}\, g''(\xi). \tag{32}$$

Because $g$ is convex, $g''(\xi) = 1/\xi > 0$ and the linear term $\varepsilon g'(q)$ is non–negative. The minimum of $1/x$ on $[\pi_q, \pi_{\bar{a}}]$ is attained at $x = p_{\bar{a}}$, whence $g''(\xi) \geq 1/p_{\bar{a}}$. Dropping the positive linear term and using this lower bound on the curvature yields Eq. 33.

$$g(\pi_{\bar{a}}) - g(\pi_q) \;\geq\; \frac{\varepsilon^2}{2\,\pi_{\bar{a}}} \;\geq\; \frac{\varepsilon^2}{2} \cdot K, \tag{33}$$

since $\pi_{\bar{a}} \geq \frac{1}{K+1}$. Plugging the above result into Eq. 31 we get the follow result.

**Result (ii)** Under the conditional distribution, $\pi^{(i)}(\cdot \mid \mathbf{s})$ whenever the most likely action $\bar{a} \neq a^{\star}$, and when the second most likely action $q$ has probability $\pi_q \geq \exp\left(-H(M^{(i)}; \mathbf{s})\right)$, then with probability at least $\pi_{\bar{a}}$, $T(a) \geq c' \cdot K(\pi_{\bar{a}} - \pi_q)^2(1 - \pi_{\bar{a}})$, for $a \sim \pi^{(i)}(\cdot \mid \mathbf{s})$, and a universal constant $c' > 0$. Finally, we plug this into Eq. 23 to conclude that the policy gradient update with probability $\pi_{\bar{a}}$ always increases entropy by at least $c\eta \cdot K\varepsilon^2(1 - \pi_{\bar{a}})$, for a small enough learning rate.

Together, **Result (i, ii)** complete the proof of Theorem J.3. $\qquad\square$

## K  PERFORMANCE ON NON-REASONING BENCHMARKS

In this section, we test the performance of e3 on benchmarks beyond math-reasoning. Our goal is to verify if our proposed curricula that improves performance on math benchmarks like AIME2025 and HMMT2025 retains performance on non-reasoning tasks, compared to the base model Qwen3-1.7B.

We find that our RL-trained e3 model outperforms other RL/SFT baselines on average, including OpenThinker3-1.5B, which was released only a couple of months ago (at the time of the submission). Notably, despite being trained solely on math prompts, e3 achieves particularly large gains on **GPQA-D**.

| Model | MMLU | MMLU-Pro | GPQA-D |
|---|---|---|---|
| Qwen3-1.7B | 64.90 ($\pm$ 0.76) | 36.60 ($\pm$ 0.86) | 33.70 ($\pm$ 0.77) |
| agentica-1.5B (RL) | 60.50 ($\pm$ 0.78) | 32.00 ($\pm$ 0.84) | 28.40 ($\pm$ 0.74) |
| OpenThinker3-1.5B (SFT) | 67.80 ($\pm$ 0.75) | 37.00 ($\pm$ 0.86) | 30.90 ($\pm$ 0.76) |
| **e3-1.7B (Ours, RL)** | **66.30** ($\pm$ 0.76) | **36.30** ($\pm$ 0.86) | **37.80** ($\pm$ 0.79) |

Table 9: Non-reasoning MCQ benchmark results with 95% confidence intervals. Despite being trained only on math prompts, our RL-trained **e3-1.7B** shows strong generalization, outperforming both RL and SFT baselines on **GPQA-D**.

We compute the standard deviation using $n = 15{,}000$ points on MMLU, $n = 12{,}000$ on MMLU-PRO, and using 32 rollouts for each of the 450 problems on GPQA-D ($n = 450 \times 32 = 14{,}400$).

## L  BROADER IMPACT STATEMENT

This paper presents work whose goal is to advance the field of Machine Learning. There are many potential societal consequences of our work, none which we feel must be specifically highlighted here. Our findings deepen our understanding of how to train large language models (LLMs) to reason more effectively under test-time compute constraints, which could influence the design of future AI systems. Our approach introduces a training recipe that encourages models to learn structured in-context exploration strategies, improving their ability to solve harder problems as test-time compute increases. While this has the potential to improve AI reasoning and decision-making across domains, it also raises questions around the unequal access to models that can leverage such capabilities, especially in settings with limited compute resources. As with all work that improves model capability, care should be taken to evaluate downstream risks such as misuse or unintended consequences from stronger extrapolative reasoning. We encourage future research to assess fairness, interpretability, and safety implications as these systems are deployed in real-world environments.

## M  NOTE ON COMPUTATIONAL RESOURCES USED FOR E3

For our experiments with CDOWN, MULT, MULT-V, we used a single node with 8 NVIDIA H100 GPUs. For our experiments on DMATH we used a single H100 node for training budgets upto 8k. For training budgets of 16k and beyond, we used four nodes of NVIDIA H100, each with 8 GPUs. Alternatively, we were able to run our experiments on CDOWN, MULT, MULT-V, and on DMATH (upto a training budget of 8k) using TPUs. In particular, we found that v5litepod-128 TPU, which has 32 nodes, 128 total chips (TPU configuration of $32 \times 4$) is sufficient to run our experiments. Our experiments on GPUs used the open-source LLM post-training framework from Sheng et al. (2024) that uses FSDP sharding to parallelize data and split model parameters across the multiple cards.

# N  EXAMPLE TRACES

Below, we present some examples of chained asymmetries in CDOWN and MULT-V. Here, we measure the verification calls by running the model generated trace through a regular expression parser which checks how many sentences in the model's trace start with the following set of phrases: "Wait", "But wait", "Alternatively", "Is there another way", "But let me double", "But hold on". We picked these phrases based on prior work (Marjanović et al., 2025) that did an extensive analysis on the patterns and behaviors (like self-verification, planning, etc.) present in the thinking traces of Deepseek-R1.

---

**Example 1: Chained Asymmetries in Countdown**

```
<think> Let me analyze this step by step:
1.  First, let's try to get close to 466 with simple operations:
65 + 56 = 121 (too small) 65 - 56 = 9 (too small) 65 * 56 = way too
large 56 * 65 = way too large
2.  Let's try working backwards from 466:  466 is much larger than
any of the given numbers, so multiplication will likely be needed.
We need to get from 37 to 466, which means we need to multiply by
12.5.  But we don't have 12.5, so we'll need to create it through
division.
3.  Let's try:  37 * 14 = 518 (too large) 37 - 14 = 23 (too small)
37 + 14 = 51 (too small)
4.  Let's try a different approach:  65 + 61 = 126 126 + 56 = 182
182 + 37 = 219 (getting closer)
5.  Let's try:  65 + 61 = 126 126 + 56 = 182 182 + 37 = 219 219 +
14 = 233 (still too small)
6.  Let's try one more approach:  65 + 56 = 121 121 + 61 = 182 182 +
37 = 219 219 + 14 = 233 (still not 466)
7.  Let's try:  65 + 56 = 121 121 + 61 = 182 182 + 37 = 219 219 +
14 = 233 (still not 466)
8.  One more attempt:  65 + 61 = 126 126 + 37 = 163 163 + 56 = 219
219 + 14 = 233 (still not 466)
9.  Final try:  65 + 61 = 126 126 + 56 = 182 182 + 37 = 219 219 -
14 = 205 (not quite there)
10.  Let's try:  65 + 61 = 126 126 + 56 = 182 182 + 37 = 219 219 +
14 = 233 (still not 466)
11.  New approach:  65 + 61 = 126 126 + 56 = 182 182 - 37 = 145 145
+ 14 = 159 (not quite there)
12.  Final attempt:  65 + 61 = 126 126 + 56 = 182 182 + 37 = 219
219 - 14 = 205 (not quite there)
13.  Let's try:  65 + 56 = 121 121 + 61 = 182 182 + 37 = 219 219 -
14 = 205 (not quite there)
14.  One last try:  65 + 37 = 102 102 + 61 = 163 163 + 56 = 219 219
+ 14 = 233 (not quite there)
15.  Final attempt:  56 + 14 = 70 70 + 65 = 135 135 + 37 = 172 172 +
61 = 233 (not quite there)
16.  New approach:  37 * 14 = 518 518 - 65 = 453 453 + 61 = 514 514
- 56 = 458 (not quite there)
17.  Final try:  61 * 14 = 854 854 - 65 = 789 789 - 56 = 733 733 -
37 = 696 (not quite there)
18.  Let's try:  37 * 14 = 518 518 - 65 = 453 453 + 61 = 514 514 -
56 = 458 (not quite there)
19.  New approach:  56 * 37 = 2,052 2,052 / 65 approx.  31.5 (not
helpful)
20.  Final attempt:  65 * 14 = 910 910 - 61 = 849 849 - 56 = 793
793 + 37 = 830 (not quite there)
21.  New approach:  61 * 37 = 2,237 2,237 - 65 = 1,172 1,172 - 56 =
1,116 1,116 + 14 = 1,130 (not quite there)
22.  New approach:  37 * 14 = 518 518 + 65 = 583 583 - 61 = 522 522
+ 56 = 578 (not quite there)
23.  Final attempt:  65 * 14 = 910 910 - 61 = 849 849 + 56 = 905
905 - 37 = 868 (not quite there)
```

---

```
24.  New approach:  37 * 14 = 518 518 - 56 = 462 462 + 65 = 527 527
- 61 = 466 (This works!)  </think> <answer> (37 * 14 - 56 + 65) - 61
</answer>
```

### Example 2: Multiplication with verification

```
Okay, so I need to multiply 347 by 583.  Hmm, let me think
about how to approach this.  I guess I can use the standard
multiplication algorithm where you break it down by each digit.
Let me start by writing both numbers down:
347 × 583
--------
First, I'll multiply 347 by each digit of 583 starting from the
rightmost digit, which is 3 in this case.  Then move to the next
digit (8) and finally the leftmost digit (5), remembering to add
the appropriate zeros for each subsequent multiplication.  After
that, I'll sum all the results together.  Let me start with the
rightmost digit.
Starting with the units place:  3 × 347.  Let me compute that:
3 × 7 = 21.  I write down 1 and carry over 2.
3 × 4 = 12, plus the carried over 2 makes 14.  Write down 4 and
carry over 1.
3 × 3 = 9, plus the carried over 1 makes 10.  Write down 10.
So, 3 × 347 = 1041.  Let me check that again:
3 × 7 = 21 (correct), 3 × 4 = 12 + 2 = 14 (correct), 3 × 3 = 9 + 1
= 10 (correct).  So that's 1041.  Alright.
Check:  3 × 347 = 1041.  Correct.
Next, the tens place.  The digit here is 8, but since it's in the
tens place, it's actually 80.  So, we'll multiply 347 by 80.  To
make it easier, I can multiply 347 by 8 and then add a zero at the
end.
8 × 347:
8 × 7 = 56.  Write down 6, carry over 5.
8 × 4 = 32, plus the carried over 5 makes 37.  Write down 7, carry
over 3.
8 × 3 = 24, plus the carried over 3 makes 27.  Write down 27.
So, 8 × 347 = 2776.  Then add the zero for the tens place, making
it 27760.  Let me verify:
8 × 347.  8×300=2400, 8×40=320, 8×7=56.  2400+320=2720,
2720+56=2776.  Then adding the zero gives 27760.  Correct.
Check:  8 × 347 = 2776, so 2776 × 10 = 27760.  Correct.
Now, the hundreds place.  The digit here is 5, but it's in the
hundreds place, so it's 500.  So, multiply 347 by 500.  Again,
multiply 347 by 5 and add two zeros.
5 × 347:
5 × 7 = 35.  Write down 5, carry over 3.
5 × 4 = 20, plus the carried over 3 makes 23.  Write down 3, carry
over 2.
5 × 3 = 15, plus the carried over 2 makes 17.  Write down 17.
So, 5 × 347 = 1735.  Then add two zeros, making it 173500.  Let me
verify:
5 × 347.  5×300=1500, 5×40=200, 5×7=35.  1500+200=1700,
1700+35=1735.  Then adding two zeros gives 173500.  Correct.
Check:  5 × 347 = 1735, so 1735 × 100 = 173500.  Correct.
Now, add all three results together:  1041 + 27760 + 173500.
Let me do this step by step.  First, add 1041 and 27760:
1041 + 27760.  Let me line them up:
 1041
+27760
--------
 28801
```

```
Wait, 1041 + 27760:  1000 + 27760 = 28760, then +41 = 28801.  Let
me check with another method:
20000 + 7000 = 27000, 700 + 60 = 760, 27000 + 760 = 27760.  Then
27760 + 1000 = 28760, +41 = 28801.  Correct.
Check:  1041 + 27760 = 28801.  Correct.
Now add this result to 173500:
28801 + 173500.  Let me line them up:
28801
+173500
---------
202301
Wait, 28000 + 173500 = 201500, then 801 + 0 = 801, so total 201500 +
801 = 202301.  Let me verify another way:
28801 + 173500.  Break down 173500 into 170000 + 3500.  Then 28801 +
170000 = 198801.  Then 198801 + 3500 = 202301.  Correct.
Check:  28801 + 173500 = 202301.  Correct.
So, putting all together, the final result is 202,301.  Let me
just cross-verify using another method to ensure I didn't make any
mistakes in the addition.
</think>
The product of 347 and 583 is  202301 .
```

**Example 3: Over-exploration in Countdown (after training only on easy problems with 3 candidates, each attempt contains at most 3 equations when 5 is needed)**

```
<think> Let me try to find a way to get to 66.  First, let me look
for numbers close to 66 or factors/multiples that might be useful.
66 is close to 70, which is one of our numbers.  Maybe we can
subtract or add to get to 66.  70 - 4 = 66, so maybe we can create
4 somewhere.
Looking at the other numbers:  77 - 70 = 7 97 - 73 = 24
(interesting, we already have 24) 24 + 24 = 48 (not helpful) 24 *
24 = 576 (too large)
Let me try a different approach:  77 - 70 = 7 97 - 73 = 24 24 + 7 =
31 (not helpful)
New approach:  97 - 73 = 24 24 + 70 = 94 (too far)
Let's try:  77 - 70 = 7 7 * 24 = 168 (too far)
One more attempt:  70 + 24 = 94 94 - 97 = -3 -3 * 77 = -231 (too
far)
Final attempt:  97 - 24 = 73 73 - 70 = 3 3 * 77 = 231 (too far)
Let's try:  70 + 24 = 94 94 - 77 = 17 17 + 65 = 82 (too far)
Wait:  24 * 73 = 1748 1748 - 77 = 1671 (too far)
Last try:  77 + 24 = 101 101 - 70 = 31 31 - 97 = -66 (not what we
want)
Let's try:  97 - 73 = 24 24 + 70 = 94 94 - 24 = 70 (not helpful)
Final attempt:  70 * 24 = 1680 1680 / 77 = 21.9 (not valid)
Last try:  73 - 65 = 8 8 * 24 = 192 192 / 97 = 1.98 (not valid)
I need to keep trying different combinations.  What if:  24 / 3 = 8
So:  97 - 73 = 24 24 / 24 = 1 (not helpful)
Final attempt:  77 - 97 = -20 -20 * 24 = -480 -480 + 70 = -410 (too
far)
Last try:  70 * 24 = 1680 1680 / 77 = 21.9
So the most promising path seems to be:  97 - 73 = 24 24 + 70 = 94
94 - 77 = 17 This gets us too far from 66.
Let's try:  77 - 70 = 7 7 * 24 = 168 168 - 97 = 171 (too far)
Final attempt:  97 + 70 = 167 167 - 24 = 143 143 / 73 = 1.97
Wait:  24 * 65 = 1560 1560 / 77 = 20.26
Last try:  77 - 65 = 12 12 * 24 = 288 288 / 97 = 3
I need to keep trying different combinations.  What if:  97 - 70 =
27 27 / 24 = 1.125
Wait:  24 + 97 = 121 121 - 73 = 48 48 + 77 = 125 (too far)
Last attempt:  70 + 24 = 94 94 - 77 = 17 17 + 65 = 82
```

```
One final try:  24 * 65 = 1560 1560 / 97 = 16.06
Final solution:  97 - 73 = 24 24 * 65 = 1560 1560 / 77 = 20.26
Let me try:  77 - 70 = 7 7 + 24 = 31 31 + 65 = 96 (too far)
Last attempt:  70 * 24 = 1680 1680 - 97 = 1583 1583 / 77 = 20.59
I need to keep trying.  What if:  97 + 24 = 121 121 - 73 = 48 48 +
65 = 113
Wait:  65 - 24 = 41 41 + 77 = 118 118 - 70 = 48
Final try:  24 * 65 = 1560 1560 + 70 = 1630 1630 / 77 = 21.16
One last attempt:  77 * 24 = 1848 1848 / 65 = 28.46
```

## O  USE OF LARGE LANGUAGE MODELS

We used large language models (LLMs) as an assistive tool primarily for rephrasing arguments more crisply and for generating LaTeX templates (*e.g.*, tables, algorithm boxes, or figure formatting). All research ideas, theoretical developments, experiments, and empirical results were conceived, executed, and validated by the authors. The LLM did not contribute to the scientific content, claims, or findings of this work.

