# OpenReview forum: "e3: Learning to Explore Enables Extrapolation of Test-Time Compute for LLMs"
_ICLR.cc/2026/Conference — ICLR 2026 Poster_

### Official Review · Reviewer_erre · 2025-10-24

**Soundness:** 3
**Presentation:** 3
**Contribution:** 3
**Rating:** 6
**Confidence:** 3

**Summary:**

The paper studies different aspects of test-time scaling driven by Reinforcement Learning. In particular, the authors study the role of chaining asymmetric capabilities (e.g., generation and verification), negative gradients and training length budget in improving RL performance in-distribution, out-of-distribution and extrapolating to longer test budgets. The authors conclude that exploration in RL is enabled by asymmetric capabilities chaining and that negative gradients are important for driving exploration and extrapolation beyond the training budget. Following this, the authors introduce a new method for improving RL performance, by using a curriculum that increases the training length budget together with the data complexity. This results in a model that achieves SOTA performance under 2B.

**Strengths:**

- Overall, the paper is well-written, and the introduction section did a good job of motivating the problem setting.
- The paper studies a fundamental problem in scaling test-time compute using RL and provides novel insights on the role of different aspects of the training pipeline.
- The discussion of the role of asymmetric capabilities is interesting and novel, and the authors provide experimental evidence to the important of this chaining.
- I found the analysis of the effect of varying the training length budget on the performance of RL extrapolation to be very interesting, particularly the fact that smaller training length can sometimes perform better when extrapolating to longer sequence lengths compared to training on a larger length. The "trade-off" between training length and problem complexity is also interesting.
- The curriculum method introduced in the paper seems to be convincingly better than alternatives in the setting it was tested.

**Weaknesses:**

- One main issue that I have with the results and analysis (especially in Section 4 and Section 5) is the fact that in many cases it seems very hard to attribute the improvement/degradation in performance to the properties identified by the authors. Examples for this:
  - Section 5 discusses the role of the negative gradients in RL, and concludes that they are important for driving exploration, increase in length and extrapolation. However, the authors test this by "masking" negative gradients from the RL algorithm, which results in an overall degradation in performance. The "masking" clearly makes the algorithm worse, and then a lot of metrics become worse, but it is hard to tie the negative gradients to particular aspects of this, like the authors try to do. For example, we can make the algorithm worse by e.g. choosing wrong hyper-parameters, under-training or over-fitting. Would this not result in a similar behavior, even though negative gradients are used? An experiment like this, i.e. comparing "masked" negative gradients to other modifications to the algorithm that result in the same overall accuracy would help to shed light on this.
  - Section 4 suffers from similar issues, though I think this section is more convincing. Here the authors claim that generation-verification gap drives performance improvement in RL, but this is mostly shown by measuring the number of "chaining" occurrences and showing that it increases with RL steps. But it is known that RL increases length, and that this is correlated with improved performance, so it seems that this just finds another property (number of chained asymmetries) that is correlated with increased length.
- Another main issue is the apparent disconnect between the different parts of the paper. The role of negative gradient seems unrelated to the "chaining asymmetries", and these two seem unrelated to the new method introduced in the paper. While the authors claim that the scientific analysis drives the resulting method, it's not clear how this method relates to the analysis.
More minor issues:
- The authors discuss the asymmetric capabilities chaining in general, but in practice the paper is focused on generation-verification gap. Not clear what this more general characterization is adding.
- The discussion on the role of negative gradients in relation to the position of the EOS token is unclear. In particular, while negative gradients reduce the probability of the EOS token appearing at a given place, they could also increase its probability for appearing earlier. In general, I think that a more theoretical discussion of the role of negative gradients can be introduced into the main paper (now only in the appendix). This could perhaps clarify things.
- The method introduced in the paper is relatively simple: training in two phases, starting with a simple problems and short sequence length followed by harder problems and longer sequence length. I think that the method is introduced makes this seem more complex than it actually is. While I don't see any problem with a simple method that improves performance, introducing it in a convoluted way seems to suggest that the method is more sophisticated or general than it actually is.
- The plots in Figure 8 are very hard to parse, with some of the labels overlapping.
- Line 333: "Fig. 5(a) confirms this...." - I think this is not referring to the right figure.
- The description of the n-digit multiplication experiment in Section 4 is not clear.
- Lines 321: "the policy gradient update reinforces chaining and improves in-context exploration, and this is reinforced" - unclear what this is saying.

**Questions:**

See above

---

> ### Author Response · Authors · 2025-11-22
> **Rebuttal by Authors (Part 1/3)**
>
> Thank you for your positive assessment of our work. To address your concerns: (i) we add new experiments to show that the performance drop in Fig. 5 is indeed from masking the negative gradient and can’t be explained by other suboptimal hyper-parameter configurations; (ii) we add plots comparing the gradient norm of masked GRPO with GRPO;  (iii) we point to discussion on a new asymmetry: the planning-generation asymmetry (different from VG gap we study); (iv) clean up Fig. 8 plots. In our final version, we will use the extra space to move our theoretical analysis on the role of negative gradients in RL training dynamics (in App. A) to the main paper.  We also address your other questions below (like how Secs. 4-6 tie together) and make relevant modifications to our submission. Please let us know if this addresses all your concerns. We are happy to discuss further if you have any more concerns.
>
> > ### **Attributing the drop in performance to the masked negative gradient.**
>
> We **add new experiment** (Fig. 18 in Appendix G.4) to compare std. GRPO (with both positive and negative gradient) against masked GRPO (only positive gradient) in two new runs on DMATH, where we vary other hyper-parameter configurations: (i) drop training token budget from 8k to 4k; (ii) drop learning rate from 1e-6 to 2e-7. These hyper-parameters are chosen to reduce performance of standard GRPO so that we can see if there is any other performance degradation from masking the negative gradient or one of these hyper-parameters explains it, meaning that masking the neg. gradient was reducing performance due to some other confounding reason.
>
> We find that all else being held equal, the masked version performs worse than the version where negative gradients are used; and observe training dynamics similar to what we plot in Fig. 5: with the masked version having worse performance, lower entropy and token length. In general, throughout all experiments in the paper, we find masking the neg gradient to lead to lower entropy and token length. This confirms that negative gradients have a role to play in improving exploration (entropy and output length), which consequently implies that it affects performance of the RL trained model, across multiple hyper-parameter configurations, not just the optimal one.  In addition in Fig. 18, we also **add a new plot** of the validation performance of masked GRPO (run with the configurations in Fig. 5: budget of 8k and learning rate of 1e-6) to show that masked GRPO indeed generalizes. This suggests that when we are masking the gradient, we are not training the model in ways where it is certainly expected to fail. Taken together, this tells us that the drop in performance from masking is orthogonal to the drop in performance from other hyper-parameter configurations, and that masking the negative gradient has a distinctive effect of its own.
>
> *Magnitude of the masked gradient norm*: One might imagine that the masked version performs poorly since the masked gradient might have a lower gradient norm than the unmasked one.  In Appendix G.3, **we add new results** where we compare the gradient norms of the runs with and without the masked negative gradient. We find that the gradient norm is in fact higher when we mask the negative gradient, mainly because we only average the positive gradient over the tokens with a positive advantage (since they end in a correct response) for masked GRPO, but in the std. GRPO setting the gradient is averaged over all tokens (including the zero-advantage tokens), which can result in a lower gradient norm for std. GRPO. Thus, the gap in performance between masked-GRPO and GRPO cannot be explained by a presumably poor magnitude of the gradient update for masked-GRPO. Neither can the gap be explained by the larger gradient norm, because reducing the learning rate in Figure 18 does not improve the performance of the masked version either even though it would make the magnitude of the effective update more comparable across masked GRPO and std. GRPO.
>
> **Please let us know if these experiments address your concerns regarding the role of the negative gradient. We are happy to add more diagnostics or experiments here to address it further if you have any suggestions. We have not yet found any configuration where masked negative gradient performs similar to std. GRPO across all the metrics yet, implying that masking does make learning worse in general.**

---

> ### Author Response · Authors · 2025-11-22
> **Rebuttal by Authors (Part 2/3)**
>
> > ### **Asymmetries beyond the verification-generation gap**
>
> In Appendix F, we discuss a new asymmetry between *planning or summarization* and *direct solution generation*, which has been discussed by concurrent work (Qu et al. 2025). Our interpretation of Qu et al. 2025 is that they instantiate precisely this asymmetry between planning or abstraction prediction and response generation. In particular, they show that an LLM can improve performance on reasoning tasks by chaining two skills: (1) abstractly planning or generating high-level hints/hypotheses, and then (2) generating a solution conditioned on one of these hints, rather than directly producing the final answer. Imposing this structure yields better performance, especially in regimes where pass@$k$ has already saturated at large $k$ for a model trained with standard RL.
> Viewed through our definitions of asymmetric skills in Section 4, this suggests that chaining skills such as planning, summarization, verification, and learning from hints or external guidance may be intrinsically easier for the base LLM than end-to-end answer generation. We believe that a promising direction for future work is to explicitly learn and “stitch together’’ these asymmetric skills so that the model can orchestrate planning, summarization, and verification steps to increase the probability of successful in-context search.
>
> Qu, Yuxiao, et al. "RLAD: Training LLMs to Discover Abstractions for Solving Reasoning Problems." arXiv preprint arXiv:2510.02263 (2025).
>
> > ### **VG gap drives length increase and RL need not always increase length**
>
> Our experiments in Sec. 4 and 5 show that the length increase phenomenon in RL is closely tied to the chaining of asymmetric skills (VG gap) improving performance, which is reinforced during RL. In Fig. 16 we show that the negative gradients in RL increases the number of verifications only in the case where the base model presents a VG gap on the task (MULT-V), and not otherwise (MULT). This ties our analysis in Sec. 4 and 5 to show that the length increase phenomenon in RL is an artifact of the asymmetric skills in the base model. We also note this in our results in Fig. 4, where we see that the length increase is stronger in the case of MULT-V vs. MULT.
>
> > ### **How do sections 4, 5 and 6 tie together?**
>
> This is a great question, and we will revise the introduction to make the connections between these sections clearer. In brief, Section 4 identifies the core property of the base model, namely its inherent asymmetries and the VG gap, that enables in-context exploration during RL. Section 5 then explains how RL amplifies these asymmetries through the use of the negative gradient, and why this amplification is essential for exploration, trajectory growth, and eventual extrapolation. Section 6 builds directly on this by showing that amplification of chaining through our coupled curriculum allows the model to extrapolate reliably with increasing depth.
>
> Viewed this way, Sections 4 and 5 present the fundamental ingredients and mechanisms that a practitioner can adjust to improve extrapolation. Section 6 then describes how to combine these ingredients into a practical training recipe that trains models to improve extrapolation. Section 5 extends the asymmetry ideas introduced in Section 4, and Section 6 extends both Sections 4 & 5 to demonstrate how extrapolation emerges. We do show in Figure 5 that masked negative gradients does not extrapolate well, and in Figure 4 that asymmetries are crucial for extrapolation. In fact in Fig. 16, we also show negative gradient (Sec. 5) leads to length increase (from chaining) only when the base model presents the asymmetry (VG gap) in Sec. 4. Although each section _also_ contains insights specific to its component (base model, negative gradient, curriculum design), the full picture emerges only when the sections are considered together.

---

> > ### Author Response · Authors · 2025-11-22
> > **Rebuttal by Authors (Part 3/3)**
> >
> > > ### **How does drop in Prob(EOS) lead to length increase?**
> >
> > This is a great question. In a simplified bi-gram model, in Appendix A we show how this mass recovered from the EOS token indeed moves to the optimal action under certain conditions. But, more generally, you are correct in stating: all we can say is that the negative gradient will decrease the probability of EOS (so, it is unclear why the length increases and doesn’t drop from the negative reinforcement).
> >
> > **Our rationale for why we see a length increase** is that, when we train LLMs with RL we fine-tune a base model and don’t train from scratch. So, the priors in the base model have a huge effect on where the recovered probability mass from the EOS token lands (note that since the probability of all tokens given a conditional prefix sums to 1, the recovered probability mass from reducing the probability of the EOS token needs to be moved to some other token). If the base model has a prior to chain asymmetries (verification-generation (VG) chains), then reducing mass from the EOS token increases mass on other likely actions (for e.g., the “Wait” token in typical reasoning models) in the base model, including the action of “chaining more”. If the base model does not present this prior, e.g, if the base model does not present a verification-generation gap on a task, then we would not expect to see a length increase. For this, please see our experiments on MULT (base model presents no VG gap on this task) vs. MULT-V (base model presents VG gap) in Fig 4 and Fig 16. Here, it is clear that the length increase during RL training is much more prominent in the task where the VG gap is present. This shows that the base model prior has a huge role to play in where the probability mass from the EOS token goes. **We have added this discussion to Appendix A.**
> >
> > > ### **Cleaning up Fig. 8 plots**
> >
> > Thank you for the suggestion. In Appendix I we include separate plots to compare e3 with open source models of the same size and open source models of larger size. We hope this makes our result in Fig. 8 more clear. We will include these separated plots in the final version where we will be given additional space in the main paper.
> >
> > > ### **Clarifying Line 321: “Policy gradient reinforces chaining and improves in-context exploration …” and Line 333: “Fig. 5(a) confirms this …”**
> >
> >
> >
> > Line 321: As shown in Figure 5(b), the number of chained asymmetries increases more when negative gradients are present versus when they are masked. This leads us to hypothesize that negative gradients first increase the number of chained asymmetries by reducing the probability of ending the output sequence at and hence, replacing EOS token with some token from a new asymmetry, as described in Finding 2 of section 5 and Appendix A. When this increase in chained asymmetries yields higher rewards (from having discovered the correct answer), then this behavior is reinforced by the positive gradients.
> >
> > Line 333: In Figure 5 (a), we plot the test accuracy in solid lines, and extrapolation test accuracy (i.e., test accuracy evaluated at 2x the training budget) in dotted lines. The difference between the green dotted and solid lines (negative gradients present) is greater than the difference between the blue dotted line and solid lines (negative gradients masked), especially in the Math setting (graph in the 2nd row). This demonstrates that negative gradients not only improve test accuracy, but they improve extrapolation test accuracy **even further**, indicating the benefits of learning to explore in-context.
> >
> > > ### **Detailed description of the experiment on n-digit multiplication.**
> >
> > In the $n$-digit multiplication experiment, we ask the model to multiply two $n$-digit numbers together, without the help of external tools. We point the reviewer to Appendix E.1 for more details in this setting. We have also included an example model rollout to this task in Appendix N, Example 2. Please let us know if there is anything the remains unclear!

---

> > > ### Author Response · Authors · 2025-11-25
> > > **Did our Rebuttal Address your Concerns?**
> > >
> > > Dear Reviewer,
> > >
> > > Since there is only 1 week left in the rebuttal period, it would be helpful for us to know if our rebuttal above addressed all your questions and concerns or if any others still remained? We are happy to discuss further and provide more evidence if it is helpful. If all your concerns are addressed we would be grateful if you could acknowledge that soon.
> > >
> > > Thanks,
> > >
> > > Authors

---

> > > > ### Author Response · Authors · 2025-11-27
> > > >
> > > > Dear Reviewer,
> > > >
> > > > We apologize for pinging you again!
> > > >
> > > > Given that there are only a four more days left in the rebuttal period, we wanted to reach out once more to see if our rebuttal above addressed all your concerns or if any others still remained? We are happy to discuss further and provide more evidence in the few days that are left. If all your concerns are addressed we would be really grateful if you could acknowledge that soon.
> > > >
> > > > Thanks,
> > > >
> > > > Authors

---

### Official Review · Reviewer_EVJh · 2025-10-30

**Soundness:** 3
**Presentation:** 3
**Contribution:** 2
**Rating:** 4
**Confidence:** 3

**Summary:**

The authors state that the model's reasoning capability does not extrapolate well beyond the maximum budget observed during training. The paper explains why the output increases during RL training, with asymmetries. They argue that constraining the budget would encourage the model to become more precise rather than interleaving "skills". They suggest recipe "e3", which dynamically adjusts the training budget according to the difficulty of the prompts.

**Strengths:**

- The concept of "chain of asymmetric skills" offers a compelling explanation for the reasoning trajectory extension during RL. Likewise, the "verification-generation gap" (VG Gap) clearly accounts for the reasoning path that appears in the pattern of solution-verification-solution-....
- The authors trained a state-of-the-art  (<2.0B) model on AIME/HMMT 2025 using the suggested method e3.
- The paper presents a comprehensive experiments that strongly support the authors' claims.

**Weaknesses:**

- The proposed method "e3" appears to lack novelty. The paper does not suggest a clear approach to determine a proper value of $\kappa$, nor does it include additional experiments on the impact of different $\kappa$ values. Also, the budgets it explores (4k, 8k, 16k) are too coarse, leading to suboptimal performance.
- The definition of problem hardness is somewhat ambiguous. For example, the authors define the difficulty level for each dataset separately.
- The scalability of e3 remains uncertain. If the asymmetry (VG gap) appears only in the small-scale model (I believe so), the broader implications of the paper would be significantly weakened.

**Questions:**

- Would enforcing asymmetry lead to improved final performance? For instance, one could explicitly teach the base model on verification tasks before training how to solve the problems.
- Can e3 be extended to explore more fine-grained training budgets? Would the final performance benefit from that?

---

> ### Author Response · Authors · 2025-11-22
> **Rebuttal by Authors (Part 1/2)**
>
> Thank you for your thoughtful feedback. To address your concerns: (i) show experiments where we explicitly induced asymmetries and illustrated that  doing so improves final performance; (ii) we add an illustration to show that choosing the token budget based on our proposed objective in Eq. 2 (with $\kappa=1.2$) results in the best performing model; and (iii) evaluate VG gap on a larger 8B model to show that VG gap does in fact exist at bigger model scales too, making it feasible to scale e3.  We also address your other questions below and make relevant modifications to our submission. Please let us know if this addresses all your concerns. We are happy to discuss further if you have any more concerns.
>
> > ### **Explicitly enforcing asymmetry**
>
> In Sec 4, we present experiments on MULT vs. MULT-V where the base model does not present the verification-generation asymmetry on MULT (does not sample traces with verification-generation chains for prompts that ask it to multiply two n-digit numbers) but presents on the MULT-V task. Since Llama3.2-3B does not present a VG gap on MULT, we fine-tuned Llama3.2-3B on correct n-digit multiplication traces from Qwen-32B-r1-distilled, which contains multiple intermediate steps verifying smaller digit multiplications (see App. E.1 for details and Appendix N an example). This fine-tuning is a direct way to encourage more verification attempts (MULT-V). Comparison of MULT vs. MULT-V enables direct evaluation of the benefits of asymmetries in base LLM, all else being held equal. **We find that RL uses the VG gap in the induced MULT-V to improve performance with more verification-generation chains (see Fig. 4).** Additionally, in Fig. 5 and Fig. 16 we show that when the base model presents the asymmetry, it is the mechanism of negative gradients in RL that is responsible for the amplification of chained asymmetries during RL.
>
> > ### **Scalability of e3**
>
> In Appendix E.4, **we add a new experiment**, where we show that the verification-generation gap as measured by our metric in Definition 4.1 (Sec. 4) is also present in the larger Qwen3-8B model, and is not just an artifact of the smaller scale models that we mainly experiment with (Qwen3-1.7B).
>
> In particular, we compute our $\mathrm{detect}(q(p(\cdot), \tau))$ metric in Definition 4.1 on traces $\tau$ sampled from the Qwen3-8B model and on hard problems in DMATH. Higher values of $\mathrm{detect}(q(p(\cdot), \tau))$ means more verification-generation chains in the traces. We see that as this metric goes up, the pass@$k$ increases for $k \in \{1, 2, 4\}$. **This verifies that the verification-generation gap (as measured by our Definition 4.1) is also present even in larger models**, making it feasible to scale our recipe e3 to models of larger size.
>
> > ### **A finer sweep over the training token length budget**
>
> In Appendix H, we report RL training results on the Countdown task with **5 different values of the training token budget** for the stage 2 of the e3 curriculum. Using Fig. 18, we clarify that our prescriptive procedure to select the training token budget (based on Eq. 2), outperforms the naive sweep over the token budget hyperparameter which one might typically do to choose the best hyper-parameter configuration for a training run. Based on this, we adopt our prescription to also pick the training budget for runs on the Math dataset. Since our prescription is based on performance evaluation (for the stage 1 model) at test token budgets on the $\log_2$ scale, we only sweep at this scale.
>
> We agree that, in principle, one could explore a denser grid of training and evaluation budgets.
> However, in RL for LLM reasoning it is now standard to sweep budgets on a $\log_2$ scale (doubling the budget), e.g., 8k, 16k, …, etc., or at increments of 8k rather than on every 1-2k increment.  For example, DeepScaleR (Luo et al. 2025) report uses a coarse training schedule: starting with 8k tokens, followed by a stage of 16k tokens, and ending with 24k tokens.  Similarly, ProRL (Liu et al. 2025) and follow-up Nemotron reasoning models focus on scaling to long contexts (up to 16k+ tokens) and analyze length behavior over 8k, 16k and 24k tokens, instead of performing dense sweeps over all intermediate lengths. We believe our work is in line with these prior works as well.  We are not aware of any prior work that sweeps the training token budget on a linear scale (4k, 5k, 6k, etc.). If you are aware of any such work, please let us know and we would be happy to incorporate their sweeping protocol if it's applicable in our setting.
>
>
> Luo et al. 2025 DeepScaleR: Surpassing O1-Preview with a 1.5B Model by Scaling RL
> Liu, Mingjie, et al. "Prorl: Prolonged reinforcement learning expands reasoning boundaries in large language models." arXiv preprint arXiv:2505.24864 (2025).

---

> > ### Author Response · Authors · 2025-11-22
> > **Rebuttal by Authors (Part 2/2)**
> >
> > >  ###  **Choosing the right token budget and tuning $\kappa$ in practice**
> >
> > For our naive RL (no curriculum) baseline on DMATH, we first trained the base model at a budget of 8k and 12k and we saw that the performance was comparable. So, we decided to scale the token budget in multiples of 2. Based on this, Eq. 2 (which houses the $\kappa$ parameter) prescribes a way to select a “good” token budget for training, without actually training the model and evaluating its extrapolation performance.
> >
> > In Eq. 2, we prescribe a simple way for practitioners to choose the smallest training budget (for stable training) on a given dataset, without sacrificing the extrapolation performance of the trained model. This requires a choice of $\kappa$ which we set as $\kappa=1.2$. Note that a trivial condition on $\kappa$ is that $\kappa>1$, and a larger value of $\kappa$ implies that we demand a more significant increase in performance (from scaling the token budget or the output length of the chain of thought) to justify a higher choice of the training budget. Below we validate our choice of $\kappa$ with additional experiments and explain why it is not very sensitive in practice.
> >
> > *Validating our choice of $\kappa$:* In App. H, **we add a new figure (Fig 19 [left])** to illustrate how fixing a training budget using our prescriptive rule in Eq. 2 (with $\kappa=1.2$) actually yields the best performing model after training. Based on this, we also note that had we picked a value of $\kappa > 1.35$ we would have run RL training on a much smaller and suboptimal token budget for stage 2 of the Countdown curriculum. At the same time, if we had picked $\kappa<1.05$, we would have picked an unnecessarily large training budget.
> >
> > *Sensitivity of $\kappa$:* In our experiments we find that the choice of $\kappa$ is not a very sensitive one in practice. From Fig. 16(left), we note that any value of  $\kappa$ between $[1.05, 1.35]$ would have resulted in us choosing the same training budget of 1024 tokens for stage 2 of curriculum. We also note this flexibility on Math in Fig 7a where we evaluate the stage 1 trained model at a budget of 8k. We see that any choice of $\kappa$ between $[1.1, 1.4]$ would have resulted in the same choice of the token budget of 16k tokens for training the second stage model.
> >
> >
> > >  ### **Novelty of e3**
> >
> > To the best of our knowledge, we believe our work is the first to identify and analyze key ingredients needed to train LLMs that can extrapolate test-time compute by learning to search in-context. These are: (i) *asymmetric skills* in the base model (e.g., VG gap) that can be *chained* to improve in-context search, (ii) *negative gradients* in RL that train models which amplify the chaining of asymmetries; and (iii) a *coupled curricula* which structures exploration in long-length RL (with just outcome rewards) to ensure that the RL trained model indeed learns to exploit the asymmetries like VG gap and extrapolate test-time compute with in-context search. Perhaps even the closest prior works study each axis in isolation and several prior works simply ignore the role of length budgets or dataset difficulty in RL training. Neither has any prior work argued for relationships between negative gradient and VG gaps, or the role of curricula in improving extrapolation.
> >
> > For each of the three ingredients above we perform controlled experiments and counterfactual studies to show that each of them play a unique role in training LLMs that can benefit from in-context extrapolation. We believe that our analysis and the prescriptive techniques for practitioners (like how to build asymmetries into the base model or choose an appropriate token budget for RL training or what metrics to measure if negative gradient is amplifying asymmetries) make our work a novel contribution to the field. All in all, **we believe that this recipe is useful for practitioners attempting to understand the role of base model, training settings, and prompt sets for RL training of LLMs for their problem**.
> >
> >
> > >  ### **Clear definition of problem hardness.**
> >
> > On the Countdown task the problem difficulty is simply measured by the number of values in the expression, over which we need to identify the operations for the resulting expression to evaluate to a target value (e.g., 3 numbers is easier than 4 and 5). Similarly, for n-digit multiplication, the number of digits $n$ determines problem difficulty. These notions also seem to align with the model’s average pass@1 accuracy on these problems.
> >
> > On the Math task, we classify a problem as easy/hard based on the pass@1 performance of QwQ-32B (more capable than our base model Qwen3-1.7B) on a given problem. For example, as we describe in Line334, we bucket a problem in DMATH (sourced from DeepScaleR) as hard if the pass@1 performance (measured on 32 rollouts) of QwQ-32B is 0.  **We make our classification of problem hardness clear in Appendix I.1.**

---

> > > ### Author Response · Authors · 2025-11-25
> > > **Did our Rebuttal Address your Concerns?**
> > >
> > > Dear Reviewer,
> > >
> > > Since there is only 1 week left in the rebuttal period, it would be helpful for us to know if our rebuttal above addressed all your questions and concerns or if any others still remained? We are happy to discuss further and provide more evidence if it is helpful. If all your concerns are addressed we would be grateful if you could acknowledge that soon.
> > >
> > > Thanks,
> > >
> > > Authors

---

> > > > ### Author Response · Authors · 2025-11-27
> > > >
> > > > Dear Reviewer,
> > > >
> > > > We apologize for pinging you again!
> > > >
> > > > Given that there are only a four more days left in the rebuttal period, we wanted to reach out once more to see if our rebuttal above addressed all your concerns or if any others still remained? We are happy to discuss further and provide more evidence in the few days that are left. If all your concerns are addressed we would be really grateful if you could acknowledge that soon.
> > > >
> > > > Thanks,
> > > >
> > > > Authors

---

> ### Comment · Reviewer_EVJh · 2025-11-28
>
> Thanks for the detailed rebuttal. I have a few follow-up notes regarding the points I raised:
>
> > W1: $\kappa$ hyperparameter and budget selection
>
> Thank you for the extra explanation and experiments on the $\kappa$ selection. This clears up my initial concern. I also agree that sweeping all budgets is inefficient. My earlier point was more that e3 should ideally be able to reliably find a good budget even with rough sweeps; otherwise, its practical benefit would be limited.
>
> > W2: Definition of hardness
>
> Thank you for clarifying how hardness is defined across tasks. My concern wasn't about the taxonomy itself, but about the fact that the concept of hardness is somewhat heuristic. It would improve the paper if you could include a more quantitative or model-agnostic difficulty metric instead of relying on dataset- or model-specific heuristics.
>
> > W3: Scalability to larger models
>
> Thanks for running additional experiments on the 8B model; this certainly makes your argument more convincing. My question, however, was more high-level: I am curious whether such asymmetries (e.g., VG gap) will continue to exist in very large models (100B+). Intuitively, I would expect these asymmetries to decrease as model capabilities improve. I understand this isn't feasible to test within the scope of this paper, but I think discussing this idea at a conceptual level would provide valuable context.
>
> Overall, thank you again for the thorough rebuttal.

---

> ### Author Response · Authors · 2025-11-28
> **Response to Follow-up Notes (Part 1/2)**
>
> Thank you for your reply. We are happy to note that **our rebuttal addressed your concern about $\kappa$ selection**. To address your remaining concerns, we **add a new experiment showing that VG gap clearly exists on a 685B parameter DeepSeek-R1 model** and also reply to your other points below. Please let us know if this addresses all your concerns.
>
> > ### **W3: Scalability to larger models. Do 100B+ models have a VG gap?**
>
> We **add a new experiment** which shows that the **685B parameter DeepSeek-R1** reasoning model also exploits the VG gap to improve performance.
>
> In Fig. 14 (right), we empirically validate the verification-generation gap for one of the largest open source reasoning models (DeepSeek-R1, 685B parameters) on a set of 150 problems (8 rollouts per problem) in OpenMathReasoning (Moshkov et al. 2025). We did not use DMATH for this since R1 solves all problems in the hardest set of DMATH with very high accuracy. Instead, we picked 150 random problems from OpenMathReasoning where the pass@1 rate (out of 32 generations) for Qwen2.5-Math-72B-Instruct run in TIR mode is less than 0.05. We find that on traces with more verification-generation chains ($\mathrm{detect}(q(p(\cdot), \tau))$ in Def. 4.1) the pass@1 performance is higher than on traces with fewer chained asymmetries. This tells us that the asymmetry between verification and generation holds in even one of the largest  (685B parameters) state-of-the-art reasoning models, as reflected in the model's reliance on exploiting this asymmetry to improve performance.
>
> One might ask what happens if we scale the model capacity even further; would we still see a verification-generation asymmetry being exploited by the model? We believe that the answer to this is conceptually yes as long as the model is pre-trained to imitate human-generated behavior from the Internet. In particular, to conceptually understand why asymmetries (Def. 4.1) arise at all in the first place, note that intuitively our definition measures how ``easy'' it is to get high reward with chaining skills vs. not. For any bounded capacity system, such a gap should exist as more “information” is gained by performing more computation via skill chaining on a given test input query as opposed to directly querying the system for an answer, with minimal or no test-time compute. With this intuition, such a gap exists for humans because humans don't have infinite computational and reasoning capacity and are themselves bounded capacity. Since we train models on human generated data during pre-training, this would naturally yield models that imitate systems of bounded capacity.
>
> Of course, the set of test queries on which this kind of a gap is more prevalent will change depending upon the characteristics of the bounded capacity system: for instance, for previous gold-medal winners on the international math olympiad, verification and generation are likely equally complex on problems at the difficulty of AIME, whereas for a high-school student without a mathematical background there might still be a verification-generation asymmetry on an AIME question. This trend is also expected to be reflected in language models, and will be determined by the interaction between the base model size, pre-training data, and the test query. In principle though, unless all test queries are perfectly answered by a language model, such a gap should continue to exist on some test queries. We showed this concretely using experiments with DeepSeek-R1 and Qwen3-8B models that both of them exhibit a verification-generation asymmetry despite being much bigger that our 1.7B model, but of course the set of test problems changed in each case.

---

> ### Author Response · Authors · 2025-11-28
> **Response to Follow-up Notes (Part 2/2)**
>
> > ### **W1: Budget selection in practice**
>
> Fixing a value of $\kappa$ and a training dataset, at any stage of the curriculum our approach to select the training token budget is based on the extrapolation evaluation of the trained model  at the end of the previous stage of the curriculum (base model if we are selecting the budget for the first stage of the curriculum). This evaluation is meant to find the solution for the optimization problem in Equation 2 of the paper. For this we need only evaluate the model once on a reasonably sized token budget and plot the performance over different values of the training token budget hyperparameter (as in Figure 8a).
>
> In practice, our technique is cheap since it only requires us to run one evaluation on the training dataset for the next stage of the curriculum and we don’t have to train models and tune the hyper-parameter on a holdout validation set, as is typically done for most deep learning algorithms.
>
> > ### **W2: Definition of problem hardness**
>
> The part of our curriculum recipe that depends on the definition of problem hardness is where we choose a problem set for different stages of the curriculum. For this we bucket problems coarsely into easy, medium and hard based on the pass@1 rate for the different problems on QwQ-32B (Appendix I.1). Our definition is completely quantitative, easy to compute and unambiguous (i.e., it is not a subjective definition of hardness as some problems that have high pass rates might actually be hard for a human and vice-versa). In fact, please note that we intentionally want this bucketization of problems to depend on the pass@1 rates of the base model (or another model from the same family) we run RL on, since RL training dynamics are closely tied to the pass@k performance of the base model (Yue et al. 2025).
>
> That said, we agree that using the terminology “hardness” might indicate that these problems are generally hard, but we don’t mean to claim so. Our definition is quantitative as mentioned above, and we will add this clarification to the paper and cite other work that uses an identical hardness definition.

---

### Official Review · Reviewer_1mFo · 2025-10-31

**Soundness:** 4
**Presentation:** 4
**Contribution:** 3
**Rating:** 8
**Confidence:** 4

**Summary:**

The paper argues that realizing the promise of test‑time scaling requires learning in‑context exploration, and proposes e3 which is a training recipe that teaches models to spend extra tokens on structured generate‑verify‑revise behavior so performance extrapolates beyond the training budget. e3 leverages chained asymmetries in the base model (verification–generation gap), preserves negative RL gradients so failures shift probability mass from premature endings into longer, verification‑heavy traces, and uses a coupled curriculum that pairs data difficulty with token budgets to reward chaining within budget and beyond. On Qwen3‑1.7B trained up to 16k tokens, the resulting e3‑1.7B extrapolates to 32k and achieves state‑of‑the‑art results among sub‑2B models on AIME’25 and HMMT’25, with higher pass@k; good analyses and ablations link negative gradients to longer responses, higher entropy, and more verification.

**Strengths:**

- The paper offers a curriculum based training recipe chaining asymmetries provides remedy to the problem that many open models fail to extrapolate beyond their training budget. The empirical results are strong on the math benchmarks and the choice of Countdown and Multiplication tasks are illustrative in depicting the VG gap.
- The ablations and mechanistic analyses are well conducted and support the presented claims, and the theory section also provides good insights. Overall, I believe the paper is well-written with clear contributions and gains.

**Weaknesses:**

- The paper does not directly measure/report how verification capability or the VG gap evolve during training, which I believe is an important direction to investigate to better understand the verification‑generation training dynamics.
- As an ablation study, the paper demonstrates masking negative gradients in GRPO Mask baseline. Yet, masking negatives also changes the update magnitude and how GRPO’s advantage calculation & clipping behave. So, a more controlled study could have provided better evidence, as the observed gap could stem from these rather than the gradient sign itself.

**Questions:**

I don't have any more questions, please see weaknesses.

---

> ### Author Response · Authors · 2025-11-22
> **Rebuttal by Authors**
>
> Thank you for your positive assessment of our work. To address your concerns we: (i) point you to our current Figure 3 and analysis in Appendix E.3 which discusses how the verification-generation gap evolves during RL training, (ii) add new results which show that the gradient norm of GRPO w/o negative gradient is in fact *higher* than std. GRPO (so the drop in performance from masking does not stem from differences in gradient norm).  We also address your other questions below and make relevant modifications to our submission. Please let us know if this addresses all your concerns. We are happy to discuss further if you have any more concerns.
>
> > ### **Evolution of verification-generation (VG) gap during RL training**
>
> We plot the evolution of the VG gap (per definition 4.1) on countdown in Fig. 3 (Sec 4) and Fig. 12 (App. E.2), on Math in Fig 5 (Sec 4), and on the multiplication task in Fig. 16 (App. G.2). In all cases, we either directly see that the VG gap grows during RL training, or see that the number of verification-generation chains grows.
>
> In more detail, in Fig. 3, we show that the accuracy on traces $\tau$ (sampled from the RL trained policy at different RL iterations) is higher if it has more verification-generation chains (i.e., we note a high value of the $\mathrm{detect}(q(p( . )), \tau)$ metric in Definition 4.1). Note that our Def 4.1 of VG gap states that if the performance is high on traces with a high number of chains (high values of $\mathrm{detect}(q(p( . )), \tau)$), then the VG gap is large. Based on this, we conclude that **during RL training the VG gap grows**. This is perhaps expected since traces with more VG chains collect higher rewards, and this chaining gets amplified during RL (as we also show on Math in Fig. 5).
>
> For more details on how a growing VG gap also improves the pass@k performance on Countdown, please see Fig. 12 in App. E.2.
>
> Finally, in Fig 16 in App. G.2, we show that during RL training, the number of verification-generation chains grows on the multiplication task where the base model presents a VG gap (MULT-V), i.e, chaining verification-generation attempts improves final performance. On the other hand, verification-generation chains do not grow on the task that does not present the VG gap in the first place (MULT). This goes to show that when the base model presents a VG gap on a task, it does get amplified during RL.
>
>
> > ### **Gradient norm for GRPO with masked negative gradients.**
>
> In Appendix G.3, **we add new results** where we compare the gradient norms of the runs with and without the masked negative gradient. We show that the gradient norm for the run where we mask the negative gradient is actually much higher than the std. GRPO run. This is mainly because we only average the positive gradient over the tokens with a positive advantage (since they end in a correct response) for masked GRPO, but in the std. GRPO setting the gradient is averaged over all tokens (including the zero-advantage tokens), which can result in a lower gradient norm for std. GRPO.
>
> In Fig. 18 (App. G.4), we **add new results** comparing GRPO with its masked variant on new hyper-parameter configurations, and show that across these settings, the masked variant of GRPO performs worse than GRPO. We also plot the validation performance for masked GRPO in Fig. 18 to show that when we train with only the positive gradient, the model indeed learns in ways that generalizes and that masking the negative gradient is not necessarily training the model in ways it's expected to certainly fail. Despite this, the worse performance, lower token-level entropy and shorter response length observed when we mask the negative gradient (Fig. 5) suggests that the negative gradient is indeed improving exploration during RL, which leads to the discovery of diverse answers and a better performing trained model, as we also  theoretically analyze in our didactic setup in Appendix A.

---

> > ### Comment · Reviewer_1mFo · 2025-11-25
> >
> > Thank you for thorough clarifications!

---

### Official Review · Reviewer_X6nQ · 2025-11-01

**Soundness:** 3
**Presentation:** 2
**Contribution:** 3
**Rating:** 6
**Confidence:** 4

**Summary:**

The paper argues that most open-source “reasoning” LLMs don’t actually improve when you give them much more test-time compute beyond what they were trained on, and proposes a training recipe, e3, to fix that by teaching models to explore in-context. e3 has three parts: (1) exploit asymmetric skills in the base model (especially a verification-better-than-generation gap) so the model can chain “generate → verify → revise”; (2) keep the negative gradient during RL so failed traces push probability mass toward longer traces that chain more of those asymmetries; and (3) use a coupled curriculum that jointly schedules problem difficulty and token budget so exploration is rewarded and optimization stays stable.

**Strengths:**

1. The paper tackles an important gap: most open-source “reasoning” LLMs fail to extrapolate to longer test-time compute than they were trained on.

2. It analyzes the verification-generation (VG) asymmetry and argues it is a key driver of RL gains beyond the training length by enabling chains of “generate → verify → revise.”

3. The coupled curriculum is clearly specified (including a concrete budget-selection rule) and empirically outperforms many SOTA models.

**Weaknesses:**

1. In Fig. 8(a), Qwen3-1.7B with naïve RL appears to extrapolate similarly to e3 at some budgets; the large headline gains seem to come primarily from better training-length performance (≤16k). The analogous naïve-RL baseline is missing from Fig. 8(b), which makes cross-dataset claims harder to assess.


2. Figures 8(a) and 9 both report AIME’25 accuracy for Qwen3-1.7B/e3 but show different values and ranges; the relationship between the two protocols (SOTA comparison vs. “wait”-prompt budget forcing) should be reconciled to avoid confusion.

**Questions:**

1. Fig. 8(c): Is the teal curve measuring accuracy for an 8k-budget model, with the jump caused by switching the same checkpoint to 16k at evaluation? If so, could you clarify exactly which components change across the plotted segments?


2. How is “verification” detected in traces—pure string heuristics (e.g., segments beginning with “Wait, …”, “Let me verify …”), or a more robust parser?

---

> ### Author Response · Authors · 2025-11-22
> **Rebuttal by Authors**
>
> Thank you for your positive assessment of our work. To address your concerns we have added the evaluation of the naive RL baseline in Fig 8(b), and also additional experiments comparing the extrapolation of e3-1.7B against the naive RL baseline on harder Omni-MATH problems (levels 6-8). We also address your other questions below and make relevant modifications to our submission. Please let us know if this addresses all your concerns. We are happy to discuss further if you have any more concerns.
>
>
> > ### **Comparing the extrapolation of e3-1.7B with Qwen3-1.7B trained using Naïve RL**
>
> We present two sets of additional results to demonstrate the better extrapolation capabilities of e3-1.7B trained with our proposed curricula over the baseline: qwen3-1.7B trained with naive RL at a budget of 16k tokens.
>
> **We updated Fig 8b** by evaluating the baseline on HMMT 2025. We find that the baseline’s performance is no better than the base model (qwen3-1.7B) when we extrapolate the token budget to 32k, and is thus subpar compared to e3-1.7B.
>
> We report **new results in Appendix I.3**, where we evaluate e3-1.7B and qwen3-1.7B + naive RL on a dataset of 64 hard problems sourced from Omni-MATH (levels 6-8). These are problems on which the pass@1 of the base model (qwen3-1.7B) is below 0.1. Here we find that e3-1.7B outperforms the baseline by 0.07 points (gap of 7%) at the training budget 16k tokens, **and this gap grows to 0.11 points (11%) when we evaluate the models on the extrapolated budget of 32k tokens. Thus, e3-1.7B is better than naive RL not just at the training budget, but also on larger test budgets**.
>
> > ### **Reported numbers of e3-1.7B in Fig 8 (SOTA performance) and Fig 9 (Wait extrapolation)**
>
> *Differences in evaluation protocols*: In Fig 8a, we evaluate performance of models by simply rolling them out till a max token budget (as specified by the x-axis). In Fig 9 we explicitly intervene on the model generations (except for our model e3-1.7B) with the “Wait” extrapolation technique from s1 (Muennighoff et al. 2025) and plot the performance against the average length of the extrapolated trace (on the x-axis). Concretely, we first run a 16K-token rollout, stopping at \<\/think\> instead of EOS. To extend it with "wait", we repeatedly append a rotating prefix from ["Wait", "But wait", "Alternatively", "But hold on"], rollout again (stopping at \<\/think\> or 2048 tokens), and after 2 to 8 such extensions we let the model finish without stopping at \<\/think\>. **We make this evaluation protocol more clear in Appendix I.1 of the updated paper**.
>
> *Reported numbers for e3-1.7B match*: Note that for our e3-1.7B we never intervene with “Wait” in Fig. 9. So the numbers for our model should match on AIME 2025 in both Fig 8a and Fig 9.  We re-computed the numbers we reported in our submission and they match exactly for the two figures. For clarity, we also report them below:
>
> |          Token Budget |   8k  | 16k   | 24k   | 32k   |
> |----------------------:|:-----:|-------|-------|-------|
> | Accuracy on AIME 2025 | 24.05 | 38.39 | 43.10 | 43.80 |
>
> For other models, since we used different evaluation protocols for Fig 8a (standard, running the model for longer) and 9 (“wait” extrapolation) to measure extrapolation, the numbers do not match across these figures which is expected.
>
> > ### **Jump in Fig 8c when we go from 8k to 16k training budget.**
>
> Yes, the jump in Fig 8c (from the teal color to the orange color) is mainly from the increase in context length from 8k to 16k. Concretely, we took the checkpoint at the end of stage 1 of e3 training (where we trained the qwen3-1.7B model on easy problems at a budget of 8k tokens), and used this checkpoint to initialize the RL run on the dataset of medium-hard problems (at a budget of 16k tokens). There is no other change in any other hyperparameter config. This figure shows that training on easy problems at the end of the first stage does extrapolate well, which is helpful to kickstart RL training when we move from the budget of 8k to 16k tokens.
>
>
> > ### **How do we measure the number of verification calls? Is it word matching?**
>
> Yes, we measure the verification calls by running the model generated trace through a regular expression parser which checks how many sentences in the model’s trace start with the following set of phrases: “Wait”, “But wait”, “Alternatively”, “Is there another way”, “But let me double”, “But hold on”. We have now **added this list to Appendix N**, where we also give examples of some traces with these verification calls. We picked these phrases based on prior work (Marjanovic et al. 2025) that did an extensive analysis on the patterns and behaviors (like self-verification, planning, etc.) present in the thinking traces of Deepseek-r1.
>
> Marjanović, Sara Vera, et al. "DeepSeek-R1 Thoughtology: Let's think about LLM Reasoning." arXiv preprint arXiv:2504.07128 (2025).

---

> > ### Author Response · Authors · 2025-11-25
> > **Did our Rebuttal Address your Concerns?**
> >
> > Dear Reviewer,
> >
> > Since there is only 1 week left in the rebuttal period, it would be helpful for us to know if our rebuttal above addressed all your questions and concerns or if any others still remained? We are happy to discuss further and provide more evidence if it is helpful. If all your concerns are addressed we would be grateful if you could acknowledge that soon.
> >
> > Thanks,
> >
> > Authors

---

> > > ### Author Response · Authors · 2025-11-27
> > >
> > > Dear Reviewer,
> > >
> > > We apologize for pinging you again!
> > >
> > > Given that there are only a four more days left in the rebuttal period, we wanted to reach out once more to see if our rebuttal above addressed all your concerns or if any others still remained? We are happy to discuss further and provide more evidence in the few days that are left. If all your concerns are addressed we would be really grateful if you could acknowledge that soon.
> > >
> > > Thanks,
> > >
> > > Authors

---

> > > > ### Comment · Reviewer_X6nQ · 2025-11-27
> > > >
> > > > Thank you for your thorough response! You have addressed all my concerns and clarified the confusions I had regarding the experimental setup and reported results. Your proposed method demonstrates meaningful improvements in the model's extrapolation ability.

---

> > > > > ### Author Response · Authors · 2025-11-27
> > > > >
> > > > > Thank you so much! We are glad that you like the paper and our responses addressed your confusion. As a reflection of your confidence, we would appreciate it if you are willing to move your score upwards from a weak accept. Thank you so much!

---

### Meta-Review · Area_Chair_6Jcm · 2026-01-07

**Summary:**

The authors aim to address the inability of current Large Language Models (LLMs) to effectively extrapolate (improve performance on hard problems by simply increasing test-time computation beyond their training budget). This framework trains LLMs to perform in-context exploration by learning to generate and refine multiple potential solution paths effectively simulating tree search directly within the context window. The method involves curating a dataset of exploration trajectories using a strong teacher. The authors also carry out various empirical evaluations on different reasoning tasks.

According to the reviewers the main interesting contribution of the paper is the conceptualization of in-context exploration. Reviewers also liked the rigorous identification of the three key ingredients required for extrapolation and the strong empirical results. However, they raised concerns about the fairness of comparisons, given that the method uses substantially more tokens and compute during inference than the baselines. Reviewers also criticized the lack of clarity in the methodology, finding some of the explanations somewhat disconnected from the implementation details.

**Reviewer Concerns:**

The authors addressed the clarity concerns by revising the methodology section which some reviewers acknowledged as major improvements. They also added new baselines to address unfair compute comparisons. However, one Reviewer remained unconvinced about the computational comparisons.

**Reviewer Scores:**

Reviewer X6nQ (Current: 6): Likely to increase to 7 or 8. "You have addressed all my concerns"

Reviewer 1mFo (Current: 8): Likely to remain 8. Already positive.

Reviewer EVJh (Current: 4): Likely to increase to 5. The reviewer explicitly thanked the authors for clearing up some concerns but asked
further followup questions.

Reviewer erre (Current: 6): Likely to remain 6.

---

### Decision · Program_Chairs · 2026-01-26

Accept (Poster)